# Microwave assisted antibacterial action of Garcinia nanoparticles on Gram-negative bacteria

Yuqian Qiao[1,2], Yingde Xu[1], Xiangmei Liu ●[3✉], Yufeng Zheng ●[2], Bo Li ●[4], Yong Han[4], Zhaoyang Li[1], Kelvin Wai Kwok Yeung ●[5], Yanqin Liang[1], Shengli Zhu ●[1], Zhenduo Cui[1] & Shuilin Wu ●[1,2✉]

Owing to the existence of the outer membrane barrier, most antibacterial agents cannot penetrate Gram-negative bacteria and are ineffective. Here, we report a general method for narrow-spectrum antibacterial Garcinia nanoparticles that can only be effective to kill Gram-positive bacteria, to effectively eliminate Gram-negative bacteria by creating transient nanopores in bacterial outer membrane to induce drug entry under microwaves assistance. In vitro, under 15 min of microwaves irradiation, the antibacterial efficiency of Garcinia nanoparticles against *Escherichia coli* can be enhanced from 6.73% to 99.48%. In vivo, MV-assisted GNs can effectively cure mice with bacterial pneumonia. The combination of molecular dynamics simulation and experimental results reveal that the robust anti-*E. coli* effectiveness of Garcinia nanoparticles is attributed to the synergy of Garcinia nanoparticles and microwaves. This work presents a strategy for effectively treating both Gram-negative and Gram-positive bacteria co-infected pneumonia using herbal medicine nanoparticles with MV assistance as an exogenous antibacterial auxiliary.

[1] School of Materials Science & Engineering, the Key Laboratory of Advanced Ceramics and Machining Technology by the Ministry of Education of China, Tianjin University, Tianjin 300072, China. [2] School of Materials Science & Engineering, Peking University, Beijing 100871, China. [3] School of Life Science and Health Engineering, Hebei University of Technology, Xiping Avenue 5340, Beichen District, Tianjin 300401, China. [4] State Key Laboratory for Mechanical Behavior of Materials, School of Materials Science and Engineering, Xi'an Jiaotong University, Xi'an, Shanxi 710049, China. [5] Department of Orthopaedics & Traumatology, Li Ka Shing Faculty of Medicine, The University of Hong Kong, Pokfulam, Hong Kong 999077, China. ✉email: liuxiangmei1978@163.com; slwu@pku.edu.cn

Gram-negative bacteria have two cell membranes, and most small molecules are unable to traverse the outer membrane (OM) and accumulate inside the bacteria[1]. Furthermore, the OM of Gram-negative bacteria is an asymmetric structure, which consists of an extracellular leaflet made up of lipopolysaccharide (LPS) units and an inner phospholipid leaflet made of phosphatidylethanolamine (PE), phosphatidylglycerol (PG), and cardiolipin (CL)[2,3]. Similarly, many herbal medicine nanoparticles composed of a variety of small molecules are also unable to pass through OM, resulting in ineffective elimination of Gram-negative bacteria. As one kind of herbal medicine, garcinia and its related extracts have been reported to possess multiple biological activities such as anti-*Staphylococcus aureus* (*S. aureus*), anti-inflammatory, antitumoral and antilipidemic properties[4,5]. However, garcinia and its related extracts have no activity against the Gram-negative *Escherichia coli* (*E. coli*)[6].

Microwaves (MV) are electromagnetic waves consisting of an electric and a magnetic field component[7], where the electric field is very important for the wave-material (non-magnetic material) interactions. When exposed to the MV, the dipole on the material attempts to realign with the alternating electric field, which may cause material structure disorder, and meanwhile, energy is lost in the form of heat through polar molecular friction during this process[8,9]. This MV thermal effect and non-thermal effect are collectively referred to as the MV effects. Specially, OM is composed of polar molecules represented by LPS and phospholipids[3]. Based on the above mentioned, we propose a hypothesis whether MV and OM can have a strong interaction to relax the OM structure to facilitate the entry of drugs. In addition, MV has a longer wavelength than light, so it has a deeper penetrating ability to tissues, which is conducive to MV thermal-assisted bacteria-killing in deep tissues[10]. Therefore, MV may be a promising candidate for exogenous destruction of OM to assist narrow-spectrum drugs to achieve broad-spectrum antibacterial in deep tissues.

Bacterial pneumonia is a major public-health problem frequently causing death in children and immunocompromised elderly people[11]. *S. aureus* and *E. coli* are two kinds of representative Gram-positive and Gram-negative pathogenic bacteria, respectively, which are the main causes of hospital-acquired infections, particularly respiratory tract infections and ventilator-associated pneumonia[12,13]. Clinically, antibiotics is still the mainstay in the treatment of bacterial pnumonia[14,15]. With the increase in microbial resistance and the high-cost and long-period in the discovery of new antibiotics, there are fewer and fewer antibiotics available, and none of the new class of antibiotics existing for combating Gram-negative bacteria have been approved in the past 20 years[16,17]. Herbal medicine including traditional Chinese medicines with multi-targets and multi-channel action mechanisms have been attracting more and more attentions in avoiding drug resistance[18]. However, most antibacterial herbal medicines are also inactive against Gram-negative bacteria. Currently, phototherapy, an emerging antibiotic-free strategy[19,20], is not suitable for treating bacterial pneumonia due to the poor penetration depth of lights[21]. Therefore, it is urgent to develop better therapeutic strategies that can effectively treat bacterial pneumonia, especially Gram-negative bacterial infections without inducing drug resistance.

In this work, we develop a strategy of using MV, as an exogenous means, to "destroy" the asymmetric OM of *E. coli*, thereby enabling *E. coli* sensitive to narrow-spectrum drugs that can only kill Gram-positive bacteria (Fig. 1). Molecular dynamics simulation discloses that MV effect causes the formation of pore in the OM, and providing a breakthrough for the entry of nanoparticles. As a proof of the proposed theory, we prepare herbal medicine Garcinia nanoparticles (GNs), which are highly effective against

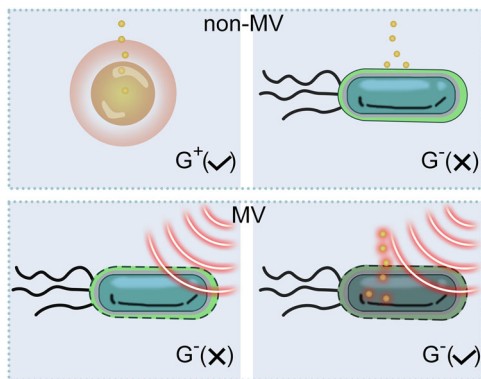

**Fig. 1 Antibacterial performance of GNs with or without MV.** GNs are sensitive to *S. aureus*, but due to the presence of outer membrane (OM) in *E. coli*, GNs have no effects on *E. coli*. After applied MV, nanopores generated in OM can induce GNs to enter the bacteria, and the subsequent synergy of GNs and its MV thermal effect plays a bactericidal effect. G+ Gram-positive bacteria, G−: Gram-negative bacteria, ✓: dead, ×: live.

both *S. aureus* and methicillin resistant *Staphylococcus aureus* (MRSA) without detectable drug resistant development over serial 30 passages, but low-effective against *E. coli*. After applying MV, the GNs could not only effectively eradicate *E. coli* in vitro, but also completely cured the bacterial pneumonia co-infected by both *E. coli* and *S. aureus* in vivo. This strategy is expected to be a new approach in the post-antibiotic era.

## Results

**The interaction between MV and the outer membrane of Gram-negative bacteria.** The influence of MV on the OM of the Gram-negative bacteria includes thermal effect and non-thermal effect[22,23]. Considering that the magnetic field force in the MV field is several orders of magnitude smaller than the electric field force[24], we ignore the influence of the magnetic field in the MV-OM interaction. In our previous work[21], we found that MV continued to irradiate for 15 min to gradually increase the temperature of the material to 55 °C without damaging normal tissues. We anticipated the interaction between MV and OM will be completed within 15 min. So, 55 °C was selected as the simulated MV thermal effect group (Ctrl). While an alternating electric field was applied at this temperature as the MV effect group (MV). The interaction between MV and *E. coli* OM was clarified by all-atom molecular dynamics simulation. Root mean square deviation (RMSD) and mean square displacement (MSD) respectively represent the relative change of the conformation of the simulated system and the root mean square displacement of all atoms in the system, which are used to judge whether the simulated system converges and the model deviation from the initial position in a solvent environment. As shown in Fig. 2a, the initial (0–10 ns) RMSD fluctuation of the OM system was due to the adjustment and adaptation of the OM to the solvent environment; after 10 ns, the RMSD of the Ctrl and MV groups became relatively stable at about 4.3 nm and 7.0 nm respectively, indicating that OM systems tended to balance under these two conditions. Meanwhile, the larger RMSD value of the MV group than the one of the Ctrl demonstrated that MV irradiation caused the looseness and deviation of the OM skeleton structure, which was also proven by the evolution of MSD of the OM as time (Fig. 2b). With the passage of the dynamic simulation time, the MSD of the MV group increased rapidly, indicating that the fluidity of the OM was increased. This is because under an alternating electric field, the charged LPS and phospholipid molecules were accelerated to move by an electric field force, resulting in a faster migration rate,

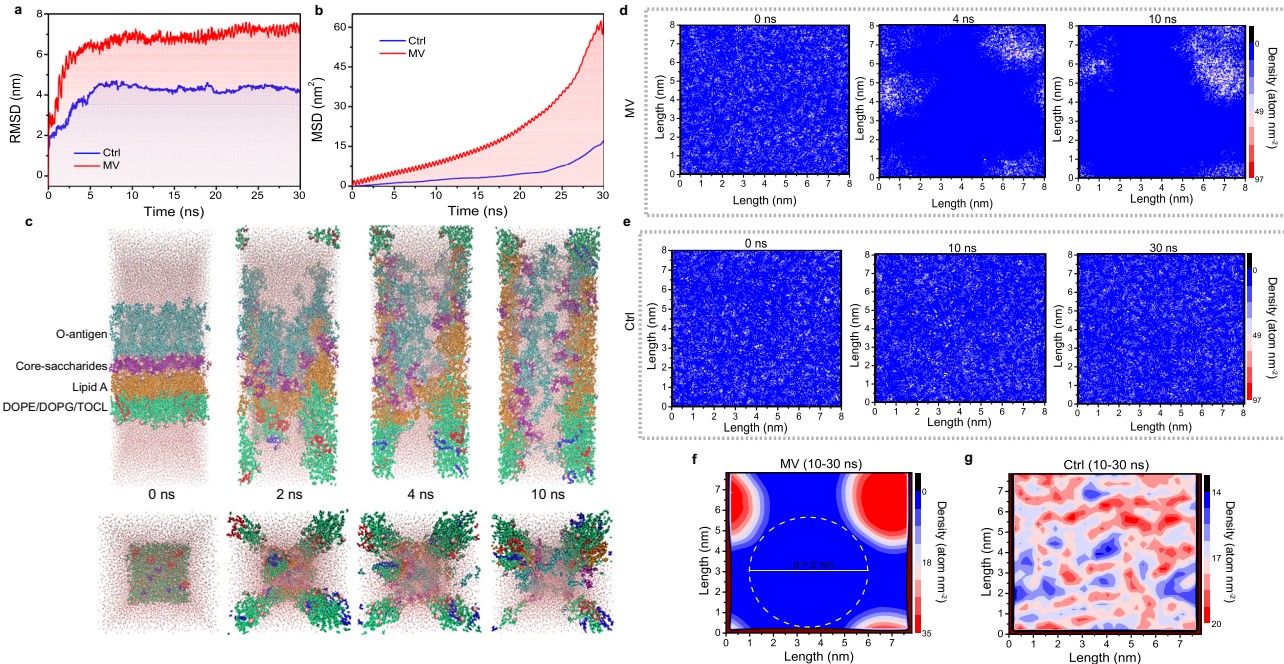

**Fig. 2 Molecular dynamics simulation of MV-induced nanopores in the OM of *E. coli*. a, b** The evolution of RMSD (**a**) and MSD (**b**) of the OM in Ctrl and MV group. **c** The conformational change of the OM after MV application at 0 ns, 2 ns, 4 ns, and 10 ns. Front view (above); Bottom view (below). **d** Two-dimensional graph of density evolution of OM after MV application at 0 ns, 4 ns, and 10 ns. **e** Two-dimensional graph of density change of OM in the group of Ctrl at 0 ns, 10 ns, and 30 ns. **f, g** Two-dimensional graph of the average density of the OM in the equilibrium phase (10–30 ns) in MV (**f**) and Ctrl (**g**) group. d, diameter. Source data are provided as a Source Data file.

*i.e.*, a larger mean square displacement suggested a stronger fluidity; this was further confirmed by the radial distribution function (Supplementary Fig. 1) and the radius of gyration (Supplementary Fig. 2) with detail explanation in Supplementary Information.

Next, the changes in the OM structure were visually analyzed. The LPS and phospholipid layers in the OM structure of the Ctrl group did not loosen or dissociate within 30 ns (Supplementary Fig. 3, Front view), and no pore structure was found in the bottom view (Supplementary Fig. 3). The structure was relatively stable within 30 ns (Supplementary Video 1 and Supplementary Video 2). As a contrast, in the MV group, with the extension of applied time of the alternating electric field, the OM structure gradually tended to loosen and dissociate, and the integrity was progressively destroyed (Fig. 2c). After 10 ns, nanopore was formed in the OM (Supplementary Video 3 and Supplementary Video 4). In order to further quantify the size of the nanopore in the OM structure, the OM molecular conformation during the dynamic simulation process was extracted and projected on the xy plane to analyze its density distribution. In the MV group (Fig. 2d), the density distribution of OM on the xy plane was relatively uniform at 0 ns. As the dynamics simulation progresses, the low-density area (blue area with a density of 0.0) gradually formed. At 10 ns, the conformational projection of the OM on the xy plane showed a large area of low-density area, indicating the gradual formation of a larger size nanopore during 0–10 ns. In contrast, in the Ctrl group, the density of OM on the xy plane within 30 ns was basically similar, with uniform distribution (Fig. 2e), and there was no obvious low-density area. Subsequently, through the density distribution analysis of all conformations in the equilibrium stage (10–30 ns) in the two groups, the average values were calculated and shown in Fig. 2f, g. In the MV group, the density distribution of OM on the xy plane was quite different during the equilibrium stage: a low-density area (blue area) with a diameter of about 5 nm (yellow dotted circle) was created to form a nanopore (Fig. 2f). In contrast, the OM in the Ctrl group on the xy plane exhibited relatively high density without low-density region, suggesting no nanopore in the Ctrl group (Fig. 2g).

Collectively, these results suggest that the MV effect caused the gradual loosening and dissociation of the OM structure as well as the self-aggregation of certain molecular chains of the OM components, thereby resulting in the formation of nanopore with a diameter of about 5 nm in the OM structure, which will help the smooth delivery of nano-medicine.

**Properties of GNs nanoparticles.** GNs were extracted from the orange gamboge resin of the garcinia hanburyi tree through hydrothermal reaction. GNs had a uniform particle size of an average size of 13.2 ± 2.2 nm (Fig. 3a), mainly consisting of nine kinds of cage-like xanthone compounds (Supplementary Fig. 4 and Supplementary Table 1). The corresponding chemical structures and proportions were respectively illustrated in Supplementary Fig. 5 and Fig. 3b, including 10-methoxygambogenic acid (1.29%), Isomorellic acid (7.25%), Gambogic acid (1.94%), Isogambogenic acid (1.53%), Gambogenic acid (28.77%), Allo-gambogic acid (0.87%), Isogambogenin (10.44%), α-Gambogic acid (43.61%), and Desoxygambogenin (4.31%). The content of α-Gambogic acid is relatively high, and its typical cage-like xanthone three-dimensional structure was shown in Fig. 3b.

The maximum UV absorption of GNs aqueous suspension was at 357 nm, and the absorbance increased with the increase of GNs concentration (Fig. 3c). Using the characteristic absorption peak of GNs at 357 nm, we studied the degradation behavior of GNs in the phosphate-buffered saline (PBS; pH 7.4). Over time, the absorbance of the GNs gradually reduced (Supplementary Fig. 6), indicating the gradual degradation of GNs in vitro. The degradability can favor GNs to be cleared from the body in a reasonable time once these herbal medicine nanoparticles have fulfilled the therapeutic functions in vivo.

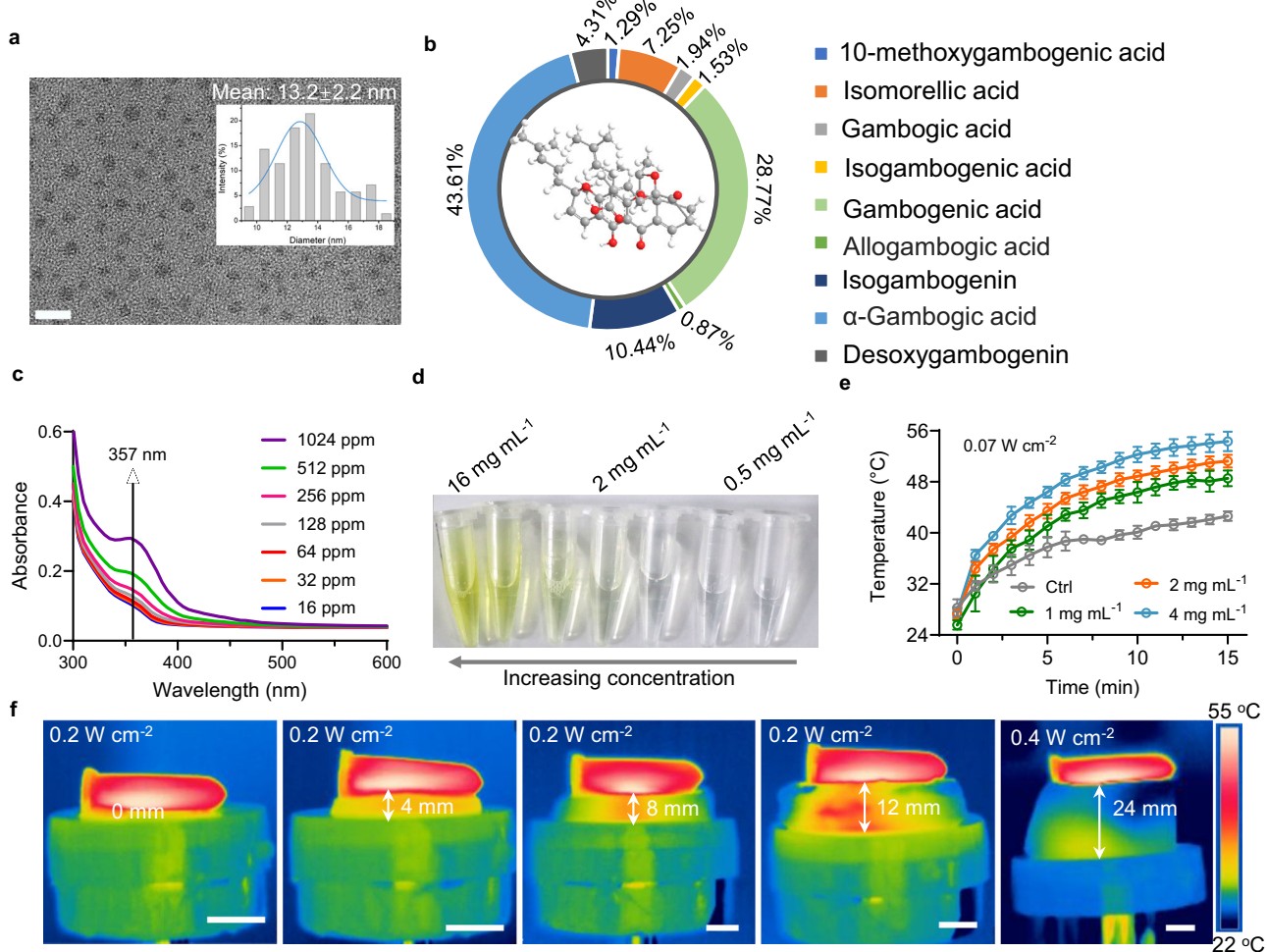

**Fig. 3 Composition and properties of extracted GNs. a** Typical TEM image of the GNs. Scale bar, 50 nm. Insert: the particle size distribution of GNs. **b** The chemical compositions of GNs and the ratio of each component. **c** UV absorbance spectra of aqueous suspensions of dispersed GNs at various concentrations, and the maximum absorbance of dispersed GNs at 357 nm. **d** Pictures of aqueous solutions with different GNs concentrations. **e** MV thermal curves of different concentrations of GNs. Data are presented as mean ± standard deviations from a representative experiment ($n = 3$ independent samples). **f** Infrared thermal images of GNs through different thicknesses (0, 4, 8, 12 mm and 24 mm) of pork tissue under MV excitation. Scale bar, 1 cm. Source data are provided as a Source Data file.

Unlike most antibiotics, GNs have a high solubility in aqueous solution. The aqueous solution of GNs was still clear even if the concentration was as high as 16 mg mL$^{-1}$ (Fig. 3d). The presence of dipoles in GNs endowed its physiological saline solution with strong MV thermal responsiveness. Under MV irradiation, the temperature of GNs solution was concentration-dependent (Fig. 3e). The temperature of 4 mg mL$^{-1}$ GNs solution rose as high as 54.3 ± 1.52 °C within 15 min MV irradiation, while under the same conditions, the control group (physiological saline) only reached the low temperatures of 42.5 ± 0.76 °C, demonstrate that GNs had a desirable MV thermal effect, which makes them a promising candidate for MV-therapy. Importantly, MV can penetrate different thicknesses of biological tissues (pork) by adjusting the MV power and irradiation time (Fig. 3f). Under 0.2 W cm$^{-2}$ MV irradiation, the time required for MV to penetrate pork with thicknesses of 0, 4, 8, and 12 mm and to bring the GNs aqueous solution to reach 55 °C were 4.46, 9.00, 9.13, and 9.46 min, respectively. It is worth noting that when the microwave power is adjusted to 0.4 W cm$^{-2}$, GNs is heated to 55 °C with 24 mm pork in only 4 min. That is to say, by adjusting the MV power and irradiation time, the GNs solution can achieve desired MV thermal performance even at 24 mm penetration

depth without causing significant heating inside the tissues (Supplementary Fig. 7), which is the key for the successful treatment of deep bacterial infections.

**GNs efficiently eradicate Gram-positive bacteria**. The representative strains *S. aureus* and MRSA of Gram-positive bacteria were used as model strains. The antibacterial tests revealed that the minimum inhibitory concentrations (MIC) of GNs against both *S. aureus* and MRSA were 64 ppm shown in Fig. 4a and Supplementary Fig. 8a, respectively. The growth curves of *S. aureus* displayed that the addition of GNs obviously inhibited the growth of *S. aureus*, and the inhibitory effect increased as the GNs concentration increased (Fig. 4b). When the GNs concentration reached MIC and 2 MIC, *S. aureus* still did not grow evidently even after 24 h of cultured. A similar phenomenon was observed for MRSA (Supplementary Fig. 8b). We also quantitatively calculated the number of *S. aureus* and MRSA colonies after co-cultured with different concentrations of GNs for 16 h. As shown in Fig. 4c, the *S. aureus* counts were obviously reduced to 0 and 2 ($10^5$ CFU mL$^{-1}$) in the 2 MIC and MIC groups with the corresponding antibacterial rate was 100% and 99.99%, respectively

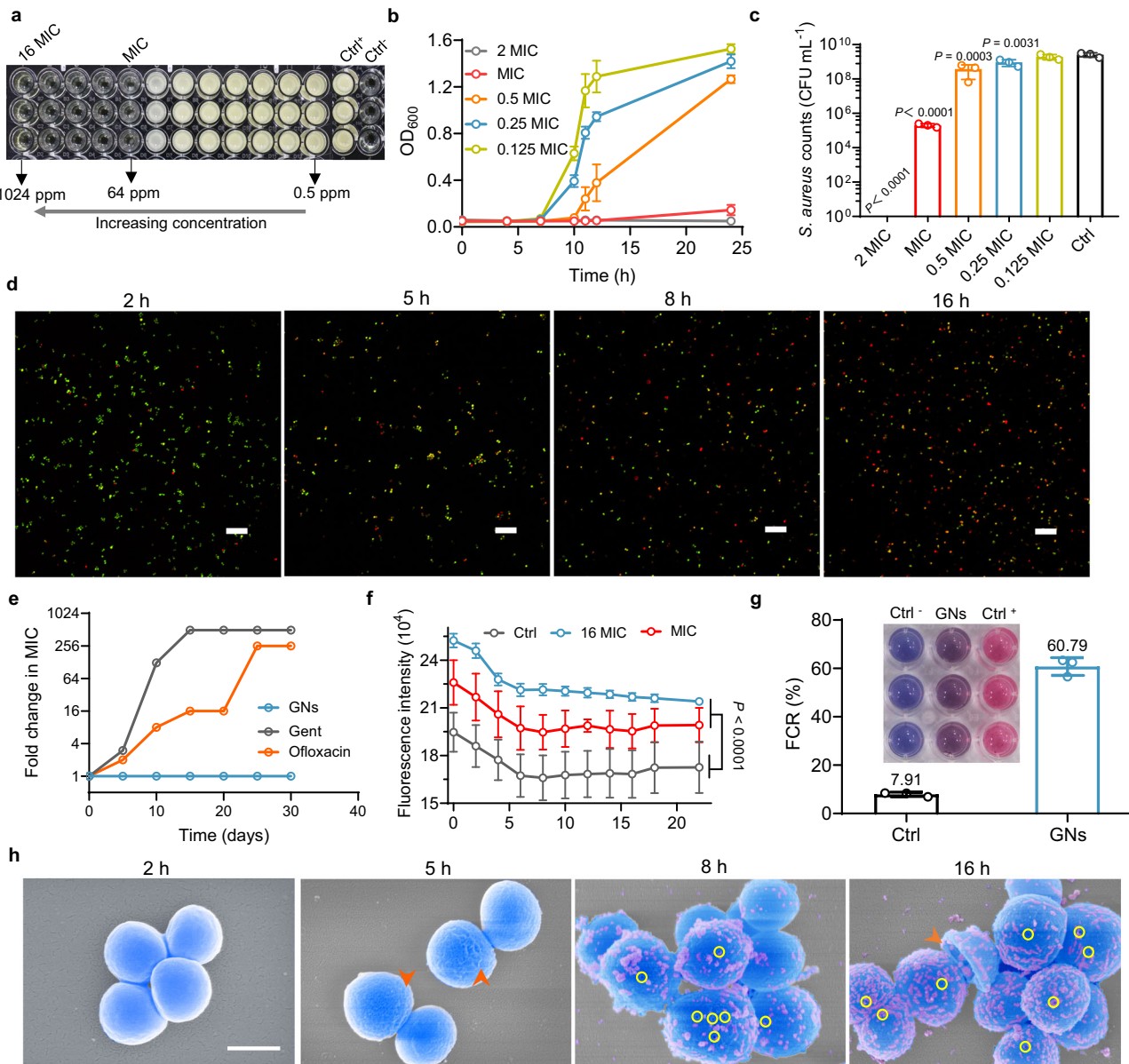

**Fig. 4 GNs efficiently eradicate Gram-positive bacteria. a** The MIC test of GNs on *S. aureus*. **b** Growth curves of *S. aureus* at different GNs concentrations. **c** *S. aureus* strain counts calculated from spread-plate assays after treatment with different concentrations of GNs. **d** Representative live-dead fluorescence staining of *S. aureus* with GNs (0.5 MIC) co-cultured for different time (2, 5, 8, and 16 h). Green fluorescence stained by SYTO9 dye indicates live bacteria, and red fluorescence stained by PI dye represents dead bacteria. Scale bar, 10 μm. **e** Monitoring of drug resistant development of *S. aureus* after exposure to GNs, Gent, and Ofloxacin at sub-MIC concentrations over serial 30 passages. **f** GNs depolarized the cytoplasmic membrane. The cytoplasmic membrane depolarization activities were determined using the membrane potential sensitive dye DiBAC4(3). **g** Interaction of GNs with the model membrane systems after incubation for 24 h. Insert: Color transitions after the interaction of GNs and DMPE/DMPG/TRCDA (1:1:3) vesicles. Ctrl⁻ and Ctrl⁺ represent the negative and the positive control group, respectively. **h** Scanning electron microscope (SEM) images representing the morphologies and structures of *S. aureus* after GNs treated different time. Scale bar, 0.5 μm. Data are presented as mean ± standard deviations from a representative experiment (*n* = 3 independent samples). *P* values were analyzed by one-way ANOVA with Dunnett's multiple comparisons post hoc test for **c** and **f**. Source data are provided as a Source Data file.

(Supplementary Fig. 9). Similarly, the MRSA counts was obviously reduced to 0 and 9.8 ($10^6$ CFU mL⁻¹) in the 2 MIC and MIC groups with the corresponding antibacterial rate was 100% and 99.62%, respectively (Supplementary Fig. 10). These results were confirmed by the spread-plate assay (Supplementary Fig. 11). It is worth noting that GNs are also highly effective in killing *S. aureus* from hospitals (Supplementary Fig. 12). *S. aureus*-killing kinetics of GNs were investigated at specific points in time (2, 5, 8, and 16 h), which showed time-dependent positive correlation characteristics (Fig. 4d). The red fluorescence in the

live-and-death staining images became stronger gradually with the increase of time, indicating that the number of killed *S. aureus* increased as the *S. aureus*-GNs interaction time increased. Notably, the drug-resistant development of *S. aureus* after exposure to the GNs, clinically used antibiotic Gentamicin (Gent) and Ofloxacin at sub-MIC concentrations over serial 30 passages was monitored (Fig. 4e). The Gent and Ofloxacin groups had 500- and 256-fold increase in drug resistant development, respectively. Nevertheless, GNs killed *S. aureus* without detectable resistant development. The reason for the long-term non-

resistance may be due to the multi-component and multi-target synergistic antibacterial effects of GNs. Next, we explored the mechanism of GNs against *S. aureus*.

Bis-(1,3-dibutylbarbituric acid) trimethine oxonol (DiBAC4(3)), a lipophilic anionic fluorescent dye, was used to detect the change in bacterial membrane potential. The increase in intracellular fluorescence intensity after adding DiBAC4(3) indicates an increase in membrane potential, that is, cytoplasmic (inner) membrane depolarization. After adding GNs, the fluorescence intensity of *S. aureus* increased significantly, indicating that GNs depolarized the bacterial membrane of *S. aureus* (Fig. 4f). Furthermore, we evaluated the propensity of GNs to associate with synthetic phospholipid groups present in bacterial inner membranes by utilizing a model membrane system comprising of PE and PG liposomes. The strong interaction between GNs and this system was demonstrated by the color transition of the model membrane system from blue to purple (Fig. 4g, insert). And, the interaction of GNs with this system increased about 7.69-fold after 24 h (Fig. 4g). Next, we observed the morphological change of *S. aureus* after GNs treatment for different time (Fig. 4h). At the beginning of the treatment (2 h), the surface of *S. aureus* was smooth. With the extension of the interaction time between GNs and *S. aureus*, the surface of *S. aureus* became slightly wrinkled (5 h), then obvious needle-like pores appeared (8 h), and finally the bacterial skeleton collapsed (16 h). This may be due to the formation of needle-like pores (indicated by yellow circle) on the surface of *S. aureus* to cause intracellular substance to flow out. Similarly, the damage of GNs treatment for 8 h to *S. aureus* could be clearly seen from the transmission electron microscopy (TEM) images (Supplementary Fig. 13). The kinetics of GNs inner membrane depolarization, the interaction with the model membrane system, and the morphological change of *S. aureus* were in good agreement with the kinetics of bacterial cell death observed in bacterial live/dead staining in Fig. 4d, indicating that the bacterial death was related to the formation of needle-like ruptures in the peptidoglycan layer and the depolarization of the inner membrane.

**MV assists GNs to kill Gram-negative bacteria.** GNs are sensitive to Gram-positive bacteria *S. aureus*, MRSA, and clinical *S. aureus* (MIC are both 64 ppm) without detectable resistance. But, GNs had insignificant activity against *E. coli* (a representative strain of Gram-negative bacteria) even the concentration reached 16 mg mL$^{-1}$ (Supplementary Fig. 14). In contrast, the low concentration of GNs (4 mg mL$^{-1}$) treatment effectively eliminated *E. coli* after 15 min MV irradiation (MV+), but MV alone showed poor antibacterial efficacy (Fig. 5a). Specifically, *E. coli* counts ($4 \times 10^8$ CFU mL$^{-1}$) in the GNs + MV group average fell to 0.005 ($P < 0.0001$, compared to Ctrl and MV), far lower than other groups (Fig. 5b), demonstrating that 99.48% *E. coli* were killed by GNs during 15 min of MV treatment (Fig. 5c). In contrast, the antibacterial rate of GNs and MV alone was 6.73% and 34.62%, respectively (Fig. 5c), far lower than that of the group of GNs + MV. Similarly, the OD$_{600}$ value of the GNs + MV group was as low as 0.007 after continuing to culture for 8 h after MV irradiation (Fig. 5d, $P < 0.0001$, compared to Ctrl, GNs, and MV), which suggesting the highly effective antibacterial efficacy against *E. coli*, which were attributed to the synergistic antibacterial effects of GNs and MV. Notably, the MV-assisted GNs treatment method also has a significant bactericidal effect on hospital-derived *E. coli* (Supplementary Fig. 15) and multi-drug resistant *Klebsiella pneumoniae* (Supplementary Fig. 16).

The combination index (CI) can be used to quantitatively evaluate the synergy of GNs and MV interactions, where CI < 1, =1, and >1 indicates synergism, additive effect, and antagonism,

respectively[25]. As shown in Supplementary Table 2, the calculated CI of GNs+MV group was 0.10942, far below 1, indicating a strong synergy between GNs and MV. And the *m* values were greater than 1 in the GNs and MV groups, indicating that the dose-effect curve was S-shaped under the two conditions, that is, with the increase of GNs or MV dose, the Fractional (F$_a$) inhibition of the two groups showed an S-shaped trend increase. D*m* is half maximal inhibitory concentration (IC$_{50}$), signifing potency. The D*m* value of GNs was 29.7184 mg mL$^{-1}$, which further revealed that GNs were not sensitive to *E. coli*. Then, we fitted the CI-F$_a$ plot of the GNs + MV antibacterial experimental data through CompuSyn software (Detailed usage are provided as a Software file) shown in Supplementary Fig. 17a, and it indicated that both GNs and MV were moderately synergistic with CI values ranging from 0.514 to 0.237 for F$_a$ = 0.5~0.97 (i.e., when *E. coli* growth was inhibited from 50 to 97%). Meanwhile, we simulated the dose-reduction index (DRI) for GNs and MV in their combination. DRI = 1, >1, and <1 indicates no dose-reduction, favorable dose-reduction, and not favorable dose-reduction, respectively[26], for GNs or MV in the combination. The favorable dose reduction was achieved (DRI > 1) as shown in the F$_a$-DRI plot (Supplementary Fig. 17b). The dose-reduction is beneficial for reducing potential toxicity of GNs in the therapeutic applications. The above results fully proved the synergistic anti-*E. coli* effect of GNs and MV.

Then, we observed the morphology of *E. coli* after different treatments. As shown in Fig. 5e, after GNs treatment, the poles of *E. coli* preferentially produced the needle-like pores (indicated by orange arrows), which may be due to the specific binding of the carboxyl groups in GNs to the amino-rich proteins of the poles of *E. coli*[27]. As a contrast, this phenomenon became more serious in the group of GNs + MV, the *E. coli* skeleton collapsed completely (indicated by blue arrows), which should be ascribed to the loss of intracellular substance. No obvious morphological changes were observed in the MV group, which may be explained as following. Although MV produced instant pores in the OM of *E. coli*, the inner membrane of *E. coli* was intact and the pores in the OM disappeared when the MV irradiation stopped. Therefore, MV alone did not induce the leakage of the inside materials in *E. coli*. Through the TEM images (Fig. 5f), we easily observed the complete Gram-negative bacterial membrane structure of GNs group. After MV was applied (GNs + MV), the inner and outer membranes were destroyed, and the intracellular substance was flowed out. These results fully indicated the synergistic effect of GNs and MV to effectively kill *E. coli*.

Furthermore, we simulated the process of GNs traversing the OM under the MV. First, the most stable structure of GNs was identified as GNs-1 by evaluating the stability parameters RDMS and Rg (Supplementary Fig. 18, the detailed discussion process was shown in supplementary information). Thus, GNs-1 was used for subsequent transmembrane simulation experiments. As shown in Fig. 5g, GNs-1 can be adsorbed in the nanopore generated by the MV, and with the extension of the simulation time, GNs-1 can migrate along the pore, and finally achieve drug delivery across the OM (Supplementary Video 5). Additionally, the density distribution of the OM-GNs-1 system before (0 ns) and after (30 ns) MV application shows that the system density distribution is uniform before MV application, and a low-density region appears in the system after MV application, which corresponds to nanopores (Fig. 5h). For ease of observation, we extracted the mean density maps of the OM and GNs-1 after MV application, respectively. As show in Fig. 5i, after MV irradiation, the OM has an obvious low-density area, and the position of GNs-1 is just in the low-density area of the OM, indicating that the GNs can counteract the barrier of OM through nanopores in the OM after MV application. In order to accelerate the penetrating dynamics of GNs-1 under MV,

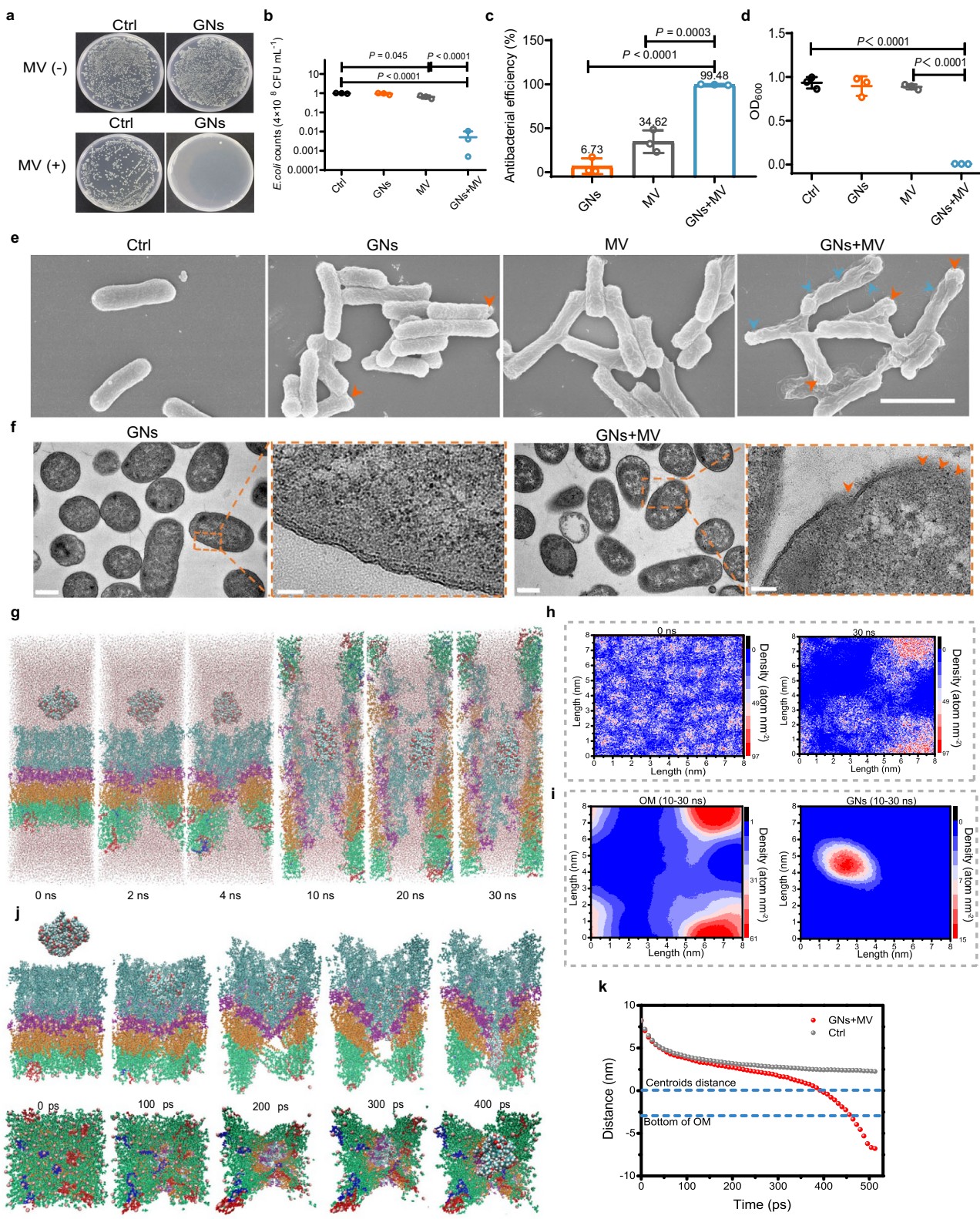

a traction force (1000 kJ mol$^{-1}$ nm$^{-2}$) was applied to GNs-1 for a tensile dynamics simulation of the OM-GNs-1 system. As shown in Fig. 5j, the OM under MV is destroyed during the kinetic process, resulting in the gradual formation of pores in the OM, so that the GNs-1 can quickly complete the transmembrane under the traction force (Supplementary Video 6). Conversely, mere traction stretching was unable to achieve the transmembrane of GNs-1 (Supplementary Fig. 19, Supplementary Video 7). This is further illustrated by the relatively large displacement of GNs-1 under MV (Fig. 5k). Therefore, GNs-1 under MV can pass through the barrier of OM and enter into cells to achieve antibacterial effect.

**Fig. 5 The efficacy of GNs and MV synergistically killing Gram-negative bacteria in vitro. a** In vitro GNs against *E. coli* under MV (MV+) or not (MV-) tested by spread plate method. **b** *E. coli* strain counts calculated from spread-plate assays after treatment with GNs under MV excitation for 15 min or not. **c** Statistics results of the antibacterial ability of *E. coli*. **d** OD$_{600}$ value of *E. coli* cultured for 8 h after different treatments. **e** SEM images representing the morphologies and structures of *E. coli* before and after different treatments. Scale bar, 2 μm. **f** Representative TEM images of *E. coli* treated with GNs under MV excitation for 15 min or not. Scale bar, 0.5 μm and 100 nm (enlarged view). **g** The conformational change of the OM-GNs-1 system after MV application at 0 ns, 2 ns, 4 ns, 10 ns, 20 ns and 30 ns. **h** Two-dimensional graph of density evolution of OM-GNs-1 system after MV application at 0 ns and 30 ns. **i** Two-dimensional graph of the average density of the OM and GNs-1 in the equilibrium phase (10–30 ns) after applied MV in OM-GNs-1 system. **j** Dynamic behavior of GNs-1 during tensile dynamics simulation under MV. **k** Displacement of GNs-1 during tensile dynamics simulation. Data are presented as mean ± standard deviations from a representative experiment (*n* = 3 independent samples for **b-d**). *P* values were analysed by one-way ANOVA with Tukey's multiple comparisons post hoc test. Source data are provided as a Source Data file.

**Mechanism of GNs collaborate with MV to kill *E. coli*.** The OM permeation of *E. coli* can be examined by monitoring the change in the fluorescent properties of the 8-anilino-1-naphthalenesulfonic acid (ANS) dye[28], a probe which displays the evolution of fluorescence upon binding to hydrophobic membrane regions. As shown in Supplementary Fig. 20, compared with the group of GNs, the group of GNs + MV showed much higher fluorescence intensity, suggesting that synergy of GNs and MV induced better OM permeation. Furthermore, the OM dye FM4-64 and the nucleic acid stain SYTOX Green (dead bacteria) were employed to determine the membrane penetration degree of the bacteria with different treatments (Fig. 6a). After GNs or MV treatment alone, a small number of bacteria died and were dyed green by SYTOX Green. Meanwhile, the FM4-64 dye in the MV group was "endocytosed" to make the entire membrane system (diffusion from the OM into the cell) or even whole body (inside the bacteria) show a red fluorescent signal (indicated by blue arrows), which was caused by the increased permeability of the OM by MV. These results indicated that the OM was perturbed under the MV to bind more lipophilic dye FM4-64, but this perturbation was not lethal, which was consistent with the results observed by SEM (Fig. 5e). Under the synergistic action of GNs and MV (GNs + MV), MV induced the formation of nanopores in the OM to increase the permeability of GNs, while GNs depolarized the inner membrane and cooperated with their own high-efficiency MV thermal effect to cause the final death of a large number of bacteria.

Furthermore, in the linescan analysis (Fig. 6b), the FM4-64 (red lines) and SYTOX Green (green lines) showed good colocalization with *E. coli* marked white arrows (randomly selected direction) in Fig. 6a confirming the successful localization of *E. coli* membrane and intracellular. In addition, the FM4-64 fluorescence intensity of *E. coli* treated with MV much higher than that in the group of Ctrl. Meanwhile, the SYTOX Green fluorescence intensity of *E. coli* treated with GNs + MV was 89-fold and 65-fold higher than *E. coli* treated with GNs and MV, respectively (Fig. 6b). These results indicated that MV can increase the permeability of OM to GNs, thereby increasing the fluorescence intensity of FM4-64 and SYTOX Green. This was further confirmed by intensity surface plot. The GNs + MV-treated *E. coli* exhibited much stronger overall fluorescence signals than those in the Ctrl, GNs or MV-treated *E. coli* (Fig. 6c). These results support the hypothesis that MV induced nanopores in the OM to allow more GNs to enter the cell, then GNs depolarized the inner membrane and cooperated with their specific MV thermal performance to cause the leakage of inside materials and the final death of Gram-negative bacteria (Fig. 6d).

**In vivo eradication of *S. aureus* and *E. coli* co-infected pneumonia.** In vitro tests showed that GNs could efficiently kill *S. aureus* and *E. coli* (under the synergy of MV). Next, we used the methyl thiazolyl tetrazolium (MTT) method to evaluate the cytotoxicity of GNs at different concentrations (Supplementary

Fig. 21a) and tested their blood compatibility by mouse blood (Supplementary Fig. 21b, c). The results show that cell viability can still reach more than 80% even the concentration of GNs reached 16 MIC and will not cause hemolysis (MIC), indicating that GNs have excellent biocompatibility and blood compatibility. To evaluate the safety of GNs in vivo, the blood tests, hepatic function, renal function and histological analysis were performed. As shown in Supplementary Fig. 21d–f, no significant difference was observed in the blood routine (WBC, Lymph, Mon, Gran, RBC, HCT, MCV, MCH, RDW, and MPV), hepatic function (ALT, TP, and TBIL), and renal function (BUN, CR, and UA) analysis between the control (without surgery) and GNs groups at a given dose. These results suggest that GNs have no appreciable toxicity and are safe for in vivo application, which was further supported by the hematoxylin and eosin (H&E) results of the internal heart, liver, spleen, lung, kidney (Supplementary Fig. 21g). Meanwhile, we verified that cell proliferation will be inhibited after MV treatment, but after a short period of recovery, the cells can gradually recover and begin to proliferate (Supplementary Fig. 22).

This MV-assisted GNs therapeutic strategy was investigated for further applications in a mouse model of *S. aureus* and *E. coli* co-infected pneumonia. First, we detected that the lung temperature of the mouse after inhaling atomized GNs reached 50 °C for 5 min after MV irradiation (Supplementary Fig. 23). In order to avoid tissue damage caused by excessive temperature, we will irradiate repeatedly (5 min each time) during the treatment. As shown in Fig. 7a, in the case of mice infected with pneumonia after co-treated with GNs and MV (GNs + MV) for 7 days, their lung tissues appeared pink and spongy, with fewer focal infections, similar to the group of Gent (pneumonia treatment with gentamicin). In contrast, the lung tissues of the control group (pneumonia treatment with physiological saline) and the GNs group (pneumonia treatment with GNs) were red and mottled with obvious focal infections. Moreover, after 7 days of treatment in different groups, micro-computed tomography (Micro-CT) images of the lungs (Fig. 7b) showed: abscesses and cavities (a typical feature of *E. coli* infection) in the lung lobes of the Ctrl group, surrounded by patches and nodular shadows and blurred edges; the treatment group of GNs showed reductions patchy shadows and irregular abscess cavities; it was noted that the abscess cavities in the group of GNs + MV and Gent disappeared with only a few patches and small nodular shadows. These results demonstrate that GNs alone had a weak therapeutic effect on *S. aureus* and *E. coli* co-infected pneumonia, but MV-assisted GNs showed excellent healing effects similar to traditional antibiotics. And, the treatment temperature of mouse lungs did not reach 55 °C, which further illustrates the importance of MV and GNs synergistically against *E. coli*.

Meanwhile, as shown in the Wright-stained (Fig. 7c) samples, the numbers of neutrophils (indicated by orange arrows) and lymphocytes (indicated by bule arrows) decreased following GNs treatment, especially in the group of GNs + MV and Gent. In contrast, the obvious neutrophils and lymphocytes were observed

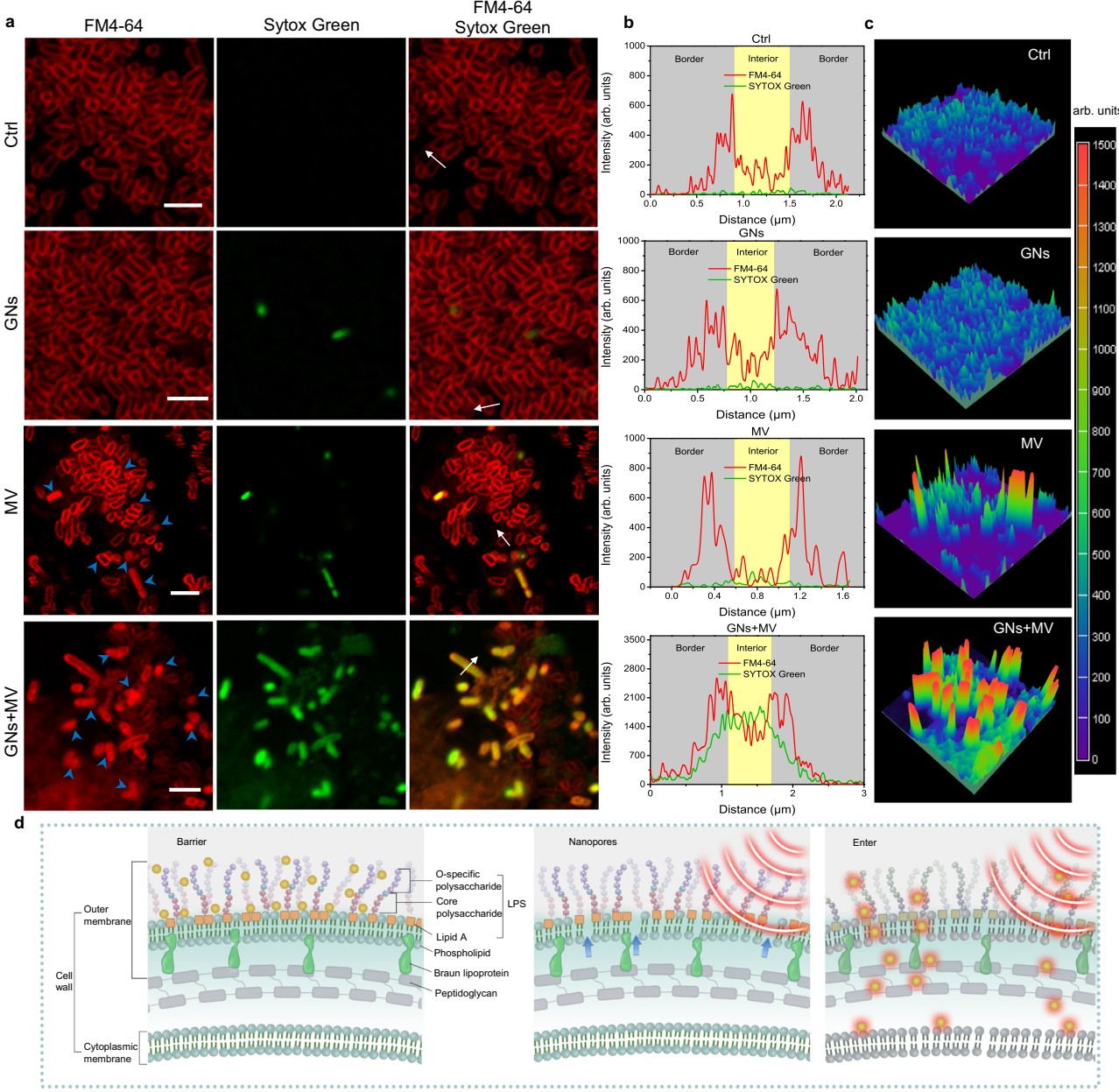

**Fig. 6 Mechanism of GNs collaboration with MV to kill _E. coli_. a** Fluorescent localization of the interactions between GNs, MV, and _E. coli_. The fluorescent dyes FM4-64 to stain the OM (red), and SYTOX Green to show membrane permeabilization (green). Scale bar, 5 μm. **b** Fluorescence intensity along the white arrows in **a**. **c** Intensity surface plot of different groups in **a**. **d** Schematic diagram of MV and GNs synergistically killing bacteria. GNs are represented by yellow spheres. Source data are provided as a Source Data file.

in blood treated with the Ctrl group. From the H&E panoramic view of the lung tissue (Fig. 7d), it was seen that the Ctrl group had a large area of pink mucus and focal infiltration of inflammatory cells, and the enlarged images clearly showed the alveolar epithelial cell hyperplasia, the alveolar septum, the alveolar wall thickening, and some alveoli shrinking, consolidation of a large number of regional organizations. After GNs treatment, the lung consolidation decreased. In contrast, the group of GNs + MV and Gent had no obvious symptoms of infections, indicating that the _S. aureus_ and _E. coli_ co-infected pneumonia had been almost eradicated.

The concentrations of interleukin 6 (IL-6, sensitive factors for diagnosing bacterial infections, Fig. 7e), granulocyte (Gran, Fig. 7f) and white blood cells (WBC) counts (Fig. 7g) were tested to evaluate the inflammatory response. After 2 days of surgery,

the IL-6 ($P < 0.0001$), Gran ($P = 0.0257$), and WBC ($P = 0.059$) levels of the GNs + MV group were significantly lower than that the Ctrl group, indicating that the bacterial infection was restrained in the GNs + MV group due to the synergistic treatment of GNs and MV. Moreover, the therapeutic effect of GNs + MV is similar to that of Gent. To quantify the antibacterial properties of GNs, we performed the colony-count assay using the harvested lung tissues at 1 day (Fig. 7h). All the treatment groups showed a reduction of _S. aureus_ and _E. coli_ colonies in the co-infected pneumonia. In addition, the GNs group has a significant therapeutic effect compared with the Ctrl group ($P = 0.0301$) in eliminate _S. aureus_, but it is not effective against _E. coli_. Notably, After MV adjuvant treatment, the number of _E. coli_ was significantly reduced compared with that of GNs group ($P = 0.0129$), which was similar to the Gent treatment

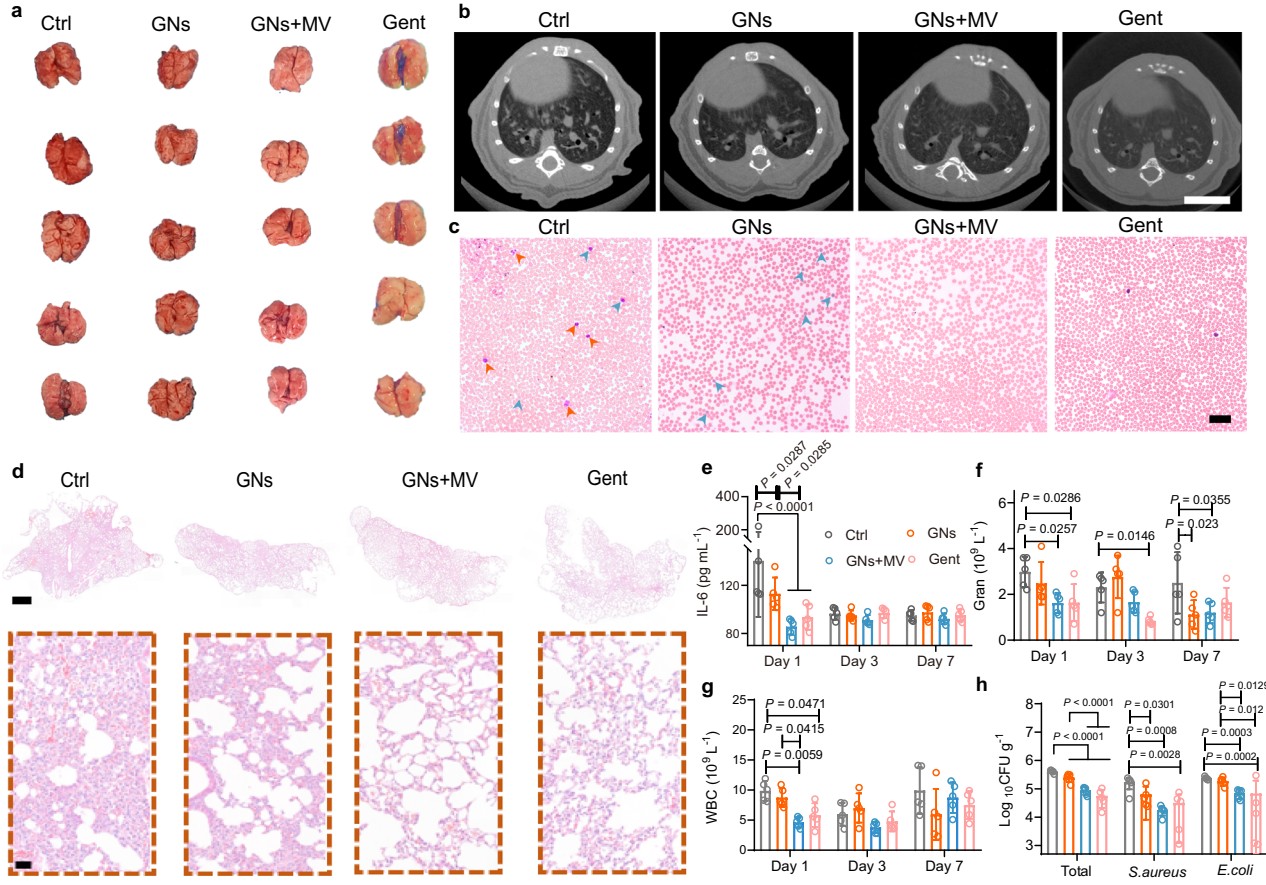

**Fig. 7 Antibacterial effects of GNs on *S. aureus* and *E. coli* co-infected pneumonia in vivo. a** Macroscopic images of lung tissue after 7 days of different treatments. **b** Micro-CT of lung in mice with co-infected pneumonia after 7 days treatment. Scale bars, 5 mm. **c** Representative Wright-stained images of blood in mice with co-infected pneumonia after one day of treatment. Scale bars, 20 μm. **d** H&E staining images of infected lung tissues after 7 days of treatment. Scale bars, 1 mm and 20 μm (enlarged view). **e-g**, IL-6 levels (**e**), amount of Gran (**f**) and amount of WBC (**g**) from 1 to 7 days in blood. **h** Bacteria counts in the infected lung after one day with different treatments. Data are presented as mean ± standard deviations from a representative experiment (*n* = 5 biologically independent samples). *P* values were analysed by two-way ANOVA with Tukey's multiple comparisons post hoc test. Source data are provided as a Source Data file.

group. Moreover, the improvement effect of GNs + MV to *S. aureus* monoinfect pneumonia (Supplementary Fig. 24) and *E. coli* monoinfect pneumonia (Supplementary Fig. 25) is similar to that of traditional antibiotics. These results proved that MV had excellent synergy effect with GNs in vivo.

Mice with severe bacteria-infected pneumonia were usually accompanied by organ injury. As shown in Supplementary Fig. 26, mice in the Ctrl group developed myocardial fibrinolysis (indicated by blackarrows) accompanied by protein mucus exudation (indicated by red arrows); a large number of irregularly shaped vacuoles (indicated by green arrows) and hepatocyte swelling (indicated by blue arrows) were seen in the liver cells; Clusters of red blood cells are gathered in the splenic sinuses (indicated by yellow arrows) with protein mucus exudation; some loop epithelial cells and loop mesenchymal cells in the medulla of the kidney tissue show watery degeneration (indicated by purple arrows). In contrast, after treatment with GNs, these abnormalities related to *S. aureus* and *E. coli* induced organ injury was partially alleviated, especially for GNs + MV group and Gent groups, which achieved better therapeutic effects. These findings indicate that GNs + MV achieves the same effect of treating pneumonia in mice as antibiotics by reducing the number of bacteria and reducing organ damage.

## Discussion

Due to the protection of the OM, the current need for an effective strategy to eliminate Gram-negative pathogens. An effective general approach that utilizes MV to generate nanopores in the OM to increase drug permeability is proposed. First, the feasibility of this strategy was verified through theoretical calculations. As a theoretical verification, GNs which are sensitive to Gram-positive bacteria (*S. aureus*, MRSA and hospital-sourced *S. aureus*) and ineffective to Gram-negative bacteria (*E. coli*), were extracted. Notably, under 15 min MV irradiation, GNs can killed 99.48% *E. coli*, 96.91% hospital-sourced *E. coli*, and 98.36% hospital-sourced multi-drug resistant *K. pneumonia*. The antibacterial mechanism is that, MV produces nanopores on OM to induce GNs to infiltrate the bacteria, and then GNs depolarize the inner membrane and cooperate with the MV thermal effect of itself to cause the leakage of intracellular substance and the final death of *E. coli*. In addition, attributable to the strong penetrability of MV, MV-assisted GNs can eradicate *S. aureus* and *E. coli* co-infected pneumonia in mice. This MV-assisted universal antibacterial method not only provides new ideas for the treatment of Gram-negative bacterial infections, but also takes an effective strategy for the treatment of deep tissue infections including bacterial infected pneumonia.

## Methods

**Materials for experiments.** All the starting materials were purchased from commercial suppliers. Orange gamboge resin of the Garcinia hanburyi tree were purchased from Bozhou Qianqian Herbal Medicine Co., Ltd. (China). Chromatographic grade acetonitrile was purchased from Kmart Chemical Technology Co., Ltd. (China). Live/Dead [TM] BacLight [TM] Bacterial Viability Kits (L7012), SYTOX [TM] Green Ready Flow [TM] Reagent (R37168), and FM [TM] 4–64 (T13320) were purchased from Thermo Fisher scientific (United States). The mice Interleukin-6 ELISA Kit (TAE-385m) was purchased from Tianjin Anoric Bio-technology Co., Ltd. (China). Starting materials were used without further purification unless otherwise noted.

**Dynamic simulation of MV-OM interaction.** Used GROMACS 5.1.2 to perform all-atom molecular dynamics simulation (MD) on OM. The initial configuration (solvent state) of OM was $8 \times 8 \times 20$ nm generated by CHARMM-GUI. The GROMOS 54a7 is a classical joint atomic force field in GROMACS. It improves the calculation rate of the all-atom force field by ignoring some atoms in the molecule (non-polar hydrogen atoms) in the calculation process and integrating the interaction with adjacent atoms that form bonds with it. That is, the calculation process is accelerated while ensuring the accuracy of the description of atomic interactions. Based on the above considerations, we used this force field for our system. Simulations were performed using GROMOS 54a7 force-field with the solvent water model of TIP3P, and the charge of the system was neutralized by calcium ions. The OM model of *E. coli* was composed of lipopolysaccharide (LPS) and phospholipids. The LPS model was composed of two O6-type antigen units, R1 core sugar, and lipid A. The phospholipid compositions were 90% 1-palmitoyl-2-oleoyl-sn-glycero-3-phosphatidylethanolamine (DOPE), 5% 1,2-dioleoyl-sn-glycero-3-phospho-rac-(1-glycerol) (DOPG) and 5% tetraoleoyl cardiolipin (TOCL1)[29]. Different colors are used to represent different components in the simulated OM: O-antigen (cyan), Core-saccharides (purple), Lipid A (brown), DOPE (green), DOPG (red), and TOCL (blue). Phosphorus atoms are represented by pink spheres.

After the system model was built, the system was first minimized with 5000 steps of energy, and then 200 ps molecular dynamics optimization was performed under NVT (canonical ensemble). Then, the system was optimized with 200 ps NPT (grand canonical ensemble) dynamics. In order to prevent any structural effects caused by the existence of the boundary, a 3 nm thick water layer was set above and below the OM. After the optimization, a 30 ns MD simulation was performed on the system under the conditions of microwave thermal effect (Ctrl) and microwave effect (MV). In all simulation calculations, the temperature coupling adopts the Nosé-Hoover method, and the pressure coupling adopts the Parrinello-Rahman method. The simulation results were analyzed with GROMACS 2018 and visualized with the Visual Molecular Dynamics (VMD). The specific calculation process of RMSD and Rg can refer to previous literature[30].

The MV field was expressed by a homogeneous time-alternating electric field as follow[31]:

$$E_{ext} = E_0 \cos(2\pi f t)(1\vec{i} + 0\vec{j} + 0\vec{k}) \tag{1}$$

where $E_0$ are the amplitude of the electric field and $f$ is the MV frequency. The applied MV fields of frequency is 2.45 GHz, which the period is ~408 ps. Considering biosafty, a small MV field is very important for actual application[32]. The actual electric field intensity in our experiment is about 2 V m$^{-1}$. However, the interaction time between the MV and the OM in our experiment was 20 min. Due to the limitation of simulation calculation ability, the actual calculation only simulates 30 ns. In order to speed up the calculation time of the interaction between the MV and OM, we increased the MV intensity to 1.0 V nm$^{-1}$. The high background electric field is typically used in MD simulations to probe poration[33].

In order to accelerate the membrane penetration process of GNs-1, the tensile dynamics simulation calculation of the OM-GNs-1 system with or without MV were carried out. During the kinetic simulation, a directional traction force was given to GNs-1 to accelerate its transmembrane behavior, and the traction force constant was set to 1000 kJ mol$^{-1}$ nm$^{-2}$. The simulation results were analyzed with Gromacs 2018 and visualized with VMD.

**Synthesis of GNs.** The GNs were prepared as follows: Orange gamboge resin of the Garcinia hanburyi tree (3.5 g) was dissolved in deionized water (35 mL) and stirred for 10 min at room temperature. It was then transferred to a 50 mL Teflon-lined stainless-steel autoclave and heated at 120 ºC for 6 h. After air cooling, the obtained product was collected by multiple centrifugation ($17,800 \times g$). And the supernatant was dialyzed using dialysis membranes (500 Da) deionized water for three days at room temperature and refreshed by deionized water every 24 h. Then, the dialyzed solution was further centrifuged for 30 min ($17,800 \times g$), and the supernatant was mixed with acetonitrile in a volume ratio of 1:1. It should be noted that acetonitrile should be slowly added dropwise to the supernatant with continuously stirring. After centrifugation of the mixed solution at ($11,400 \times g$, 30 min), the supernatant was rotarily steamed to remove the organic solvent and reconstituted with deionized water. Finally, the reconstituted aqueous solution was lyophilized to obtain GNs.

**Component analysis of GNs.** GNs (2.5 mg) was dissolved in 50% acetonitrile solution (1 mL) and filtered with a 0.22 μm filter. The resulting solution was used for Liquid chromatography-electrospray time-of-flight mass spectrometry system detection: including 1200 fast liquid chromatography (Agilent, USA), MicrOTOF-Q II electrospray-quadrupole-time-of-flight mass spectrometry (Bruker Daltonics Inc, USA). Specifically, the HPLC conditions were as follows: (1) Chromatographic: Agilent Zorbax SB- C18 (3.5 μm, 100 mm × 2.1 mm); (2) Flow rate: 0.2 mL min$^{-1}$; (3) Mobile phase A (0.1% formic acid in water) and mobile phase B (Acetonitrile) using a gradient program (B: 60% in 0–7 min, 70–90% in 7–13 min, 90% in 13–23 min, 90–55% in 23–23.1 min, and 55% in 23.1–28 min); (4) Oven temperature: 30 ℃; (5) Injection volume: 5 μL.

The MS parameters were as follows: (1) Source type: electrospray ionization (ESI); (2) Scanning method: negative ion switching scan; (3) Scan range: 50–1500 m/z; (4) Spary voltage: 4.5 kV; (5) Gas temperature, 200 ℃; (6) Data collection time: 28 min. (7) Online calibration with sodium formate standard solution. Then, the data was collected and integrated using Data Analysis 4.1 software. And the results were compared and matched with the databases (ChemSpider) and the compounds in Garcinia.

**Evaluation of GNs stability.** In the molecular dynamics process, a GROMOS 54A7 force field was applied to the GNs complexes, while water molecules were built by the TIP3P model. After the small-molecule dispersion system is established, the energy optimization of the system is carried out by the steepest descent method and the conjugate gradient method. In order to adapt the system to the simulated environment, the constant temperature ensemble equilibrium and the constant pressure ensemble equilibrium were carried out. During the equilibrium process, the temperature coupling adopts the V-rescale method, and the thermal coupling time constant is 0.1 ps. Then the GNs were simulated for 100 ns using the leapfrog algorithm, the integration step was set to 2 fs, the long-range electrostatic interaction was processed by the PME algorithm, the short-range Coulomb cutoff radius was set to 1.2 nm, and the van der Waals cutoff radius was set to 1.2 nm. The system adopts periodic boundary conditions in all directions and uses the LINCS algorithm to constrain the bond lengths of small molecules. The simulation results were tracked with Gromacs 2018 and visualized with VMD.

**Microwave thermal effect measurements.** The different weights of GNs (1 mg, 2 mg, and 4 mg) were dissolved in physiological saline (1 mL) and microwaved (2.45 GHz, 0.07 W cm$^{-2}$, Schneider Medical Equipment Co., Ltd., China) for 15 min in a 2 mL Eppendorf (EP) tube. The temperature of GNs solution was recorded every minute using an FLIR thermal camera (FLIR E50, United States). Similarly, the EP tube containing the GNs solution (1 mL, 2 mg mL$^{-1}$) was placed on biological tissues (pork) of different thicknesses (0, 4, 8, and 12 mm). After that, microwaves were applied across the biological tissues and the time required for the temperature to reach 55 ℃ is recorded.

**Minimum inhibitory concentration assays.** The MIC of GNs were defined as the lowest concentration of GNs that results in no visible bacterial growth. MICs were measured by bacterial suspension ($10^7$ CFU mL$^{-1}$) adding 2-fold dilutions of different concentrations of GNs in 96-well plates and grown with shaking at 37 ℃ (16 h). Growth curves of bacteria were monitored by measuring the optical density at 600 nm (OD$_{600}$). Assays performed by a microplate reader (SpectraMax I3MD, USA). Meanwhile, the concentrations of GNs in locations were listed (16 MIC include sixteen folds change compared to the MIC for that bacteria).

**Model membrane system.** The simulated vesicles were composed of DMPE, DMPG and TRCDA. First, DMPE, DMPG and TRCDA (mass ratio is 1:1:3) were dissolved in a mixed solution of chloroform and ethanol (volume ratio is 1:1), and then evaporated to a constant weight. Next, it was reconstituted with deionized water and sonicated for 5 min (20 W) under an ultrasonic probe, and this solution was stored at 4 ℃ overnight. Finally, the above solution was placed in a 96-well plate and added separately with PBS (Ctrl$^-$), GNs (MIC), and sodium hydroxide (Ctrl$^+$), and then irradiated at 254 nm for 30 s, resulting in the appearance of a strong blue color, due to the polymerization of the diacethylene units, and the color of damaged vesicles subsequently changed.

**In vitro antibacterial experiments.** The in vitro antibacterial activity of GNs against *S. aureus* (ATCC 25923), MRSA (CCTCC 16465), and *E. coli* (ATCC 8099) were quantitatively by the spread plate method. Unless otherwise specified, the concentration of the three kinds of bacteria was $10^7$ CFU mL$^{-1}$ diluted by overnight culturing of bacterial suspension in the antibacterial experiment.

For anti-Gram-positive bacteria test. Adding the corresponding concentration of GNs (Ctrl, 0.125 MIC, 0.25 MIC, 0.5 MIC, MIC, and 2 MIC) to the diluted bacterial solution (*S. aureus* or MRSA) cultured at 37 ℃ for 16 h. Then, diluting the co-cultured bacterial suspension by the corresponding multiple and smearing 20 μL onto the LB agar plate cultured at 37 ℃. The colonies were counted after culturing for 20 h to calculate the antibacterial ratio, which was then assessed using Eq. (2):

$$\text{Antibacterial ratio}(\%) = \frac{A - B}{A} 100\% \tag{2}$$

where A and B represent the numbers of bacteria in the control group and experimental group, respectively.

For anti-Gram-negative bacteria test. Adding GNs (4 mg mL$^{-1}$) to the diluted *E. coli* solution and then irradiated MV (0.07 Wcm$^{-2}$) for 15 min. Next, diluting the treated *E. coli* bacterial suspension and smearing 20 µL onto the LB agar plate cultured at 37 °C for colony counts.

Parameters of GN and MV synergistically killing of *E. coli*. First of all, we determined the antibacterial rates of different concentrations of GNs (mg), different powers of MV (W), and the combination of GNs and MV at different doses. Then using CompuSyn software to get the calculation results.

The morphologies of the bacteria (both *S. aureus* and *E. coli*) with different treatment were evaluated using SEM (S4800, Japan) and TEM (JEOL JEM-2100F, Japan). For SEM, the treated bacteria were soaked with glutaraldehyde (2.5%) solution for 4 h (at 4 °C) and then washed by PBS (pH = 7.0). Next, the samples were dehydrated in different concentrations (30, 50, 70, 90, and 100%, v/v) of ethanol for 15 min and air-dried before further observation. For TEM, the treated bacteria were soaked with glutaraldehyde (2.5%) solution for 24 h (at 4 °C) for fixation and then washed by PBS (pH = 7.0). Next, the fixed bacteria were post-fixed in 1% osmic acid at 4 °C for 2 h and washed with PBS. Afterwards, the fixed bacteria were dehydrated with ethanol according to the dehydration steps in the above SEM. Then, the samples were embedded in an embedding medium and cut into ultrathin sections to perform TEM observation.

**Inner membrane depolarization assay of *S. aureus*.** Diluting the *S. aureus* bacteria solution with saline to form a suspension with OD = 0.5 and adding (Bis-(1,3-Dibutylbarbituric Acid) Trimethine Oxonol) (DiBAC4(3), 5 µM). After equilibration for 30 min at 37 °C, taking 160 microliters of the above solution and adding 40 microliters of GNs (0, 5 MIC, and 80 MIC) and then the fluorescence intensity change of *S. aureus*-GNs interaction within 20 min were measured (excitation, 622 nm; emission, 670 nm).

**Outer membrane permeability assay of *E. coli*.** Diluting the *E. coli* bacteria solution with saline to form a suspension with OD = 0.5 and add ANS (10 µM). After equilibration for 30 min at 37 °C, taking 160 microliters of the above solution and adding 40 microliters of GNs (5 MIC) or after irradiation with MV (15 min) and then the fluorescence intensity between 450–550 nm was measured with excitation at 380 nm.

**Morphological staining of *E. coli*.** Diluting the *E. coli* bacteria solution with normal saline to form a suspension (10$^7$ CFU mL$^{-1}$). After different treatments, adding SYTOX Green (5 µM) with followed incubation for 4 min. Then, adding FM4-64 (5 ppm) and then incubating for 1 min (on ice). Each stained culture was then centrifuged at 6000 × g for 30 s (4 °C) and resuspended in 1/10 volume of normal saline (on ice). Taking 5 microliters of resuspension solution on the glass slide and covering it with a cover glass for photographing by laser scanning confocal microscopy (Nikon A1R+, Japan).

**Serial passaging assay to evolve resistance.** *S. aureus* grew in 0.5 MIC of the indicated medicines (GNs, Gent, and Ofloxacin) and were repeated for 30 passages, and then subsequently cultured in the absence of antibiotics, and the MIC was then remeasured. The 0.5 MICs of GNs, Gent, and Ofloxacin against *S. aureus* are 32 ppm, 12 ppm, and 2 ppm, respectively.

**In vitro cytotoxicity evaluation.** The NIH-3T3 (ATCC CRL-1658) was cultured in Dulbecco's minimum essential medium alpha (α-MEM) and supplemented with 10% (v/v) fetal bovine serum, 1% amphotericin and 1% penicillin–streptomycin and incubated at 37 °C in 95% humidity and an atmosphere containing 5% CO$_2$.

MTT assay of NIH-3T3. The various concentrations of GNs (Ctrl, 0.25 MIC, MIC, 2 MIC, 4MIC, and 16 MIC) were co-cultured with NIH-3T3 cells (10$^5$ cells/well) for three days. The Ctrl group was used the same volume of physiological saline instead of different concentrations of GNs. After incubation for three days, the medium was removed and 3-(4,5-dimethylthiazol-2-yl)-2,5-diphenyltetrazolium bromide (MTT, 0.5 mg mL$^{-1}$) was added, and then incubated for 4 h at 37 °C. Next, the supernatant was removed and 200 µL dimethyl sulfoxide was added with 15 min shaking for reconstitution. Finally, the absorption of the supernatant was monitored at 570 nm (OD$_{570}$). The cell viability (%) was calculated by comparing the absorbance values of these samples with the control.

For the MTT test method of A549 and L929 under MV irradiation, refer to the method of NIH 3T3. The difference is that after the cells adhered (after 24 h of culture), the MV was irradiated for 15 min. Other experimental steps remain unchanged.

Evaluation hemolysis of GNs. First, 2.5 mL of C57BL/6 N mice blood was diluted with 2.5 mL of PBS solution and centrifuged (176 × g) for 5 min to obtain red blood cells (RBCs). Then, the collected RBCs were washed with PBS and resuspended in PBS solution (10 mL). Different concentrations of GNs were incubated with diluted RBCs for 4 h at 37 °C (n = 3). After centrifugation (176 × g) for 5 min, the absorbance of the supernatant was measured at 405 nm.

The positive control (100% lysis) was used with 1% TritonX-100 (a kind of surfactant can destroy cell membrane) instead of GNs. The hemolysis percentage was calculated by comparing the absorbance values of these samples with the positive control.

**In vivo safety.** C57BL/6 N mice (4–6 weeks old) as the model animals were used for in vivo safety test. First, the mice were anaesthetized by isofluorane. Then, GNs (0.5 mg per mouse) were inhaled by mice from the nasal cavity by atomization. And, the control group was the mice without inhaling GNs. After 7 days of treatment, the blood routine, renal function, and hepatic function were measured. And histological analysis of major organs (heart, liver, spleen, lung, and kidney) was carried.

**Mouse bacterial infection pneumonia model and treatment.** The study was carried out in accordance with the Guide for the Care and Use of Laboratory Animals of the National Institutes of Health. The ethical aspects of the animal experiment were approved by the Animal Ethical and Welfare Committee (AEWC) of the Institute of Radiation Medicine, Chinese Academy of Medical Sciences (Approval No. IRM-DWLL-2021093) and approved by Yi Shengyuan Gene Technology (Tianjin) Co., Ltd. (Approval No. YSY-DWLL-2021035). The mice (4–6 weeks old) were randomly divided into three groups (n = 5 per group) at every preset time point: the group of control (Ctrl), the treatment group of GNs (GNs), the collaborative treatment group of GNs and MV (GNs + MV), and positive control antibiotic group (Gent).

Mice were raised under a 25 ± 2 °C (room temperature) 60–70% (humidity), and 12 h light/dark cycle conditions for 3 days. They were anaesthetized with isofluorane prior to construct an experimental model of mice pneumonia. After anesthetized, for *S. aureus* and *E. coli* co-infected pneumonia mode, 20 microliters of a mixture suspension of *S. aureus* (10$^8$ CFU) and *E. coli* (10$^8$ CFU) were sucked by a pipette into the nasal cavity of the mouse. For monoinfect pneumonia mode, replace 20 microliters of mixed bacteria suspension with single bacteria suspension. After the bacterial suspension was completely inhaled, the mouse body was kept at 90° to the operating table, and maintaining this position for 30 s to ensured that the bacterial suspension flowed into the lungs. After one day, these mice were subjected to different aerosol therapy. The Ctrl group was treated with physiological saline (5 mL per 15 mice). The GNs group was treated with GNs (7.5 mg per 15 mice), and the GNs + MV was treated with GNs (7.5 mg per 15 mice). Next, a microwave probe (8 cm in diameter, 2.45 GHz, 0.07 W cm$^{-2}$) was used to irradiate the lungs of the mouse for 5 min. After an interval of 1 min, which is to bring the temperature down to body temperature, a second irradiation (5 min) was performed. This repeated 3 times for a total of 15 min of irradiation. Multiple irradiations can avoid tissue damage caused by high temperature of the target tissue. And, the Gent group was treated with Gent (7.5 mg per 15 mice).

To evaluate the therapeutic effect of different treatments for *S. aureus* and *E. coli* co-infected pneumonia, the infected lungs specimens of the mice models were observed, photographed, and investigated using Micro CT scanner (Quantum FX) after 7 days of treatment. At a preset time, the mice were sacrificed, their blood were collected for Wright's staining (Days 1) and blood tests (IL-6, Gran, and WBC at Days 1, 3, and 7). Meanwhile, their lungs and main organs were collected for H&E staining (Days 7). Each mouse in different groups was weighed 70 mg lung tissue, added 630 microliters of PBS, and homogenized with a homogenizer (Dhyana 400DC). Homogenates used a plate coating method to determine the total bacterial counts. After incubation for 30 h at 37 °C, the golden colonies were characteristic colonies of *S. aureus*, and the remained white colonies are *E. coli* colonies. And the number of bacteria was expressed as l g CFU g$^{-1}$ for tissues.

**Statistics and reproducibility.** All the quantitative data in each experiment were evaluated and analysed by one-way or two-way analysis of variance and expressed as the mean values ± standard deviations from at least three independent experiments, followed by Dunnett's,Tukey's or Sidak's multiple comparisons post hoc test to evaluate the statistical significance of the variance. The *n.s.* present P > 0.05 and P < 0.0001 were statistically considered significance.

**Reporting summary.** Further information on research design is available in the Nature Research Reporting Summary linked to this article.

## Data availability

Source Data is provided for Fig. 2a, b, d–g, 3a, c, e, 4b, c, e-g, 5b–d, h, I, k, 6b, 7 e–h and Supplementary Fig. 1, 2, 6, 8b, 9, 10, 12c, 14b, 15b, c, 16b, c, 17, 18b, c, 20, 21a, c, d–f, 22a, c, 23d, 24c–f, 25c–f can be found in the Source Data File with this paper. All data from this study are available in the supplementary information and source data.

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

## Acknowledgements

This work is jointly supported by the China National Funds for Distinguished Young Scientists (no. 51925104 to S.W.), National Natural Science Foundation of China (Nos. 51871162 to S.W., and 52173251 to X.L.), NSFC-Guangdong Province Joint Program (Key program no. U21A2084 to X.L.), NSFC key program (No. 51631007 to Y.H.), RGC/NSFC (N_HKU725-1616 to K.W.K.Y.), as well as Hong Kong ITC (ITS/287/17, GHX/002/14SZ to K.W.K.Y.).

## Author contributions

Y.Q., X.L. and S.W. conceived and designed the concept of the experiments. Y.Q. and Y.X. performed the experiments and conducted the material characterizations. Y.Q., X.L. and S.W. analyzed the experimental data and co-wrote the manuscript. Y.Z., B.L., Y.H., Z.L., K.W.K.Y., Y.L., S.Z., Z.C., and S.W. provided important experimental insights and performed data analysis partially. All the authors discussed, commented and agree on the manuscript.

## Competing interests

The authors declare no competing interests.
