## [Peer Review File · Nature Communications]

Reviewers' Comments:

Reviewer #1:

Remarks to the Author:

The authors proposed a new and effective method to eliminate Gram-negative bacteria by the synergistic action of garcinia nanoparticles and microwave irradiation. In vivo and in vitro experiments verified that the microwave-assisted drug treatment of herbal medicine nanoparticles can cause strong inhibition on Gram-negative bacteria, and have potential therapeutic effects on Gram-38 negative and Gram-positive bacteria co-infected pneumonia. Combined with molecular dynamics simulation method, the inhibition mechanism was analyzed. This study surely could attract considerable interest. I recommend acceptance of this paper for publication after a major revision with following critical concerns.

1. In the simulations, an electric field of 1.0 V/nm was used for a very small simulation system. The actual microwave field possible used for drug delivery purpose should be considerably small compared with that used in this work. And the microwave field used in other MD simulation investigations was also considerably small. (e.g. IEEE Transactions on Microwave Theory and Techniques, 2008, 56(11): 2511-2519.). Please give an explanation.
2. As shown in Fig.2, there existed some holes in such a small system under the action of a very strong electric field. It is can be concluded that the membrane structure became looser and more porous after microwave irradiation. However, it is not reasonable to identify the holes as nanochannels. The authors should give verification and evidence of the existence of a channel.
3. Why didn't the authors perform the MD simulation for the system including membrane and nanoparticles under microwave irradiation. It would be more pertinent to this study.
4. The MARTINI force fields were developed based on the reproduction of partitioning free energies between polar and apolar phases for various chemical compounds. The addition of microwaves with rapidly changing electric fields would certainly affect the microscopic energy distributions and polarizability. Thus the implement of the force field to microwave action may need to modify or be validated. At least, the authors should have a discussion on the aspect.
5. As mentioned in the references cited by the authors (e.g. Ref.11), the microwave irradiation certainly lead to the temperature increase for the treated system. How did the authors consider this problem in the MD simulation? What is reason to perform the simulation under NVT or NPT?
6. The box size of the simulation system was set as $8 \times 8 \times 20$ nm, and a hole with a diameter of 5 nm was formed in the membrane at the end of simulation. Will the size of the simulation system affect the simulation results and conclusions?
7. In vivo experiments, the operating parameters of MW irradiation should be given, such as the microwave electric field strength/power and operating time.
8. The authors should point out how they controlled the temperature of target tissues of the mice. Where is the target tissue of the mice? The temperature was controlled at 55 °C? How to determine if the temperature was controlled at a desired value?
9. The treatment in Ref.11 was focus on MRSA-infected osteomyelitis. So, is 55 °C also the temperature needed for infected pneumonia concerned in this study? Why?
10. In Supplementary Fig.1, "the $g(r)$ decreased sharply in the long range...This suggests that some of the remote OM molecules have diffused beyond the monitoring distance". What is the monitoring distance? What is its physical implication? And what is its value?
11. In Supplementary Fig.1(b), all the values of RDF $g(r)$ at 30 ns were lower than those at 0 ns, and $g(r)$ converged to the minimum 1 at earlier time comparing with that at 0 ns. RDF denotes the ratio of the local density of average density of the system. Whether some molecules were not included in the calculation of RDF? please explain the reasons.

12. In Supplementary Fig.10, the difference in result of antibacterial test between the 0.125MIC MV group and the control group were not as large as that illustrated in Supplementary Fig.8 (26.06%), especially for those in the third line.

Reviewer #2:

Remarks to the Author:

Summary: This study entails investigating the use of Garcinia nanoparticles to eradicate Gram-negative bacteria using microwave assistance to induce nanochannels in the bacterial outer membrane to induce antibiotic entry. In vitro and in mouse models, the authors use this approach to demonstrate the use of microwave irradiation to markedly increase drug entry into cells. Strengths of this study include its use of this interesting and novel antibiotic-sparing approach to enable treatment of Gram-negative bacterial infections and inclusion of molecular, in vitro and mouse models to demonstrate effectiveness. Overall, the methods appeared to be sound for establishing the mechanism of bacterial killing by the nanoparticles. However, the manuscript could better establish clinical feasibility of this approach. For the in vivo models, it is unclear why the authors chose to investigate and only report data for *S. aureus* and *E. coli* co-infection as polymicrobial pneumonia is relatively uncommon. To establish effectiveness of this approach against Gram-negative bacterial infections, which seems to be the primary objective of this manuscript, *E. coli* monoinfection should also be tested and this data reported (even if negative) as this could suggest additional bacterial or host mechanisms at play that are required for eradication of *E. coli* in the co-infection model. The organisms tested here are lab strains so it is unclear how the approach would fare against hospital-associated pathogens which may have multiple mechanisms of antibiotic resistance and other cell wall alterations. Finally, I am not an expert in these methods, a more robust exploration of potential collateral effects on human cells may also be warranted, particularly given the time required for microwave penetration to deep tissues. There were also many errors in spelling and sentence structure and poor clarity of some sections of the manuscript that require further revision.

Specific comments:

Lines 50-51: This is perhaps meant to refer to the discovery of new drug classes e.g. from screening synthetic chemical libraries; several new antibiotics particularly with activity against carbapenem-resistant and other multidrug-resistant Gram-negatives have been approved in the last few years.

Lines 58-59: Please restate as the outer membrane presumably predates the use of antibiotics.

Lines 62-64: This sentence is unclear, please restructure. Also it would be helpful to include a brief summary of prior research specifically related to the Garcinia nanoparticles used here.

Lines 94-100: Please correct several typographical errors and fragment sentences. Please briefly state conditions in which prior testing of normal tissues was done and duration of anticipated treatment tolerance.

Lines 188-189: I don't believe heating inside tissues is reported here. To properly penetrate the human chest wall, depths >12 mm would need to be evaluated; were increasingly higher temperatures seen with treatment times >15 minutes?

Lines 224-227: Was GN given in combination with antibiotics for the duration of the treatment period? This is a bit confusing, please clarify. Please also state the concentration of GN used in comparable serial passage experiments.

Lines 356-357: Further explanation of finding of in vivo safety experiments would be helpful here. What parameters were used to compare tissue samples between the treatment and control groups? Did this follow exposure to GN only or also GN+MV?

Lines 360-362 (and similar comparisons throughout this section): Was infection with *S. aureus* or (especially) *E. coli* alone investigated for GN+MV in the in vivo models? The authors could also consider including a control group treated with standard antibiotics with activity against these two bacterial pathogens to show that improvement seen with GN+MV is similar to that seen with traditional antibiotics.

Lines 385-387: Day 1 reductions in WBCs in the treatment groups may not be meaningful, particularly given that they were higher in subsequent days. It is also unclear what effect GN

treatment itself may have on inflammatory parameters as there was no uninfected group treated with GN.

Response to Reviewer 1#

Original Comment: The authors proposed a new and effective method to eliminate Gram-negative bacteria by the synergistic action of garcinia nanoparticles and microwave irradiation. In vivo and in vitro experiments verified that the microwave-assisted drug treatment of herbal medicine nanoparticles can cause strong inhibition on Gram-negative bacteria, and have potential therapeutic effects on Gram-negative and Gram-positive bacteria co-infected pneumonia. Combined with molecular dynamics simulation method, the inhibition mechanism was analyzed. This study surely could attract considerable interest. I recommend acceptance of this paper for publication after a major revision with following critical concerns.

Reply: We express our sincere thanks to the reviewer for his/her very positive comments and valuable suggestions, which will definitely help us further improve the quality of this work.

Comment 1: In the simulations, an electric field of 1.0 V/nm was used for a very small simulation system. The actual microwave field possible used for drug delivery purpose should be considerably small compared with that used in this work. And the microwave field used in other MD simulation investigations was also considerably small. (e.g. IEEE Transactions on Microwave Theory and Techniques, 2008, 56(11): 2511-2519.). Please give an explanation.

Reply: Thank you very much for the professional suggestion. Yes, as reviewer said, a small microwave field is very important for actual application. The actual electric field intensity in our experiment is about 2 V/m, which is indeed smaller than that used in theoretical calculations. However, the interaction time between the microwave (MV) and the outer membrane (OM) in our experiment was 20 minutes. Due to the limitation of simulation calculation ability, the actual calculation only simulates 30 ns. In order to speed up the calculation time of the interaction between the MV and OM, we increased the microwave intensity to 1.0 V/nm. The high background electric field is typically used in MD simulations to probe poration (J. Song, *et.al*, Synergistic effects of local temperature enhancements on cellular responses in the context of highintensity, ultrashort electric pulses, *Med. Biol. Eng. Comput.*, vol. 49, no. 6, pp. 713–718, Jun. 2011.). It serves as an accelerated test of the pore formation process, since low electric fields would take inordinately long simulation time (J. T. Camp, *et al.*, Cell Death Induced by Subnanosecond Pulsed Electric Fields at Elevated Temperatures, *IEEE Trans. Plasma Sci.*, vol. 40, no. 10, pp. 2334-2347, Oct. 2012.).

In addition, the electric field frequency of this manuscript is 2.45 GHz, and the actual intensity is 2 V/m. The reference pointed out by the reviewer mentioned “The aim of this work is to study the carbon monoxide binding to myoglobin, considering the whole protein in water, under the exposure to a 1-GHz

EM field, to understand if microwave fields of intensities lower than 100 mV/m could alter the binding or unbinding processes". It can be concluded that the research focus of this reference is the case where the electric field strength is less than 100 mV/m. Obviously, our electric field strength is beyond the scope of the literature.

To better illustrate this point, we have modified the manuscript, as followed:

In Page **29-30**, we added the statement "...Considering biosafety, a small MV field is very important for actual application³². The actual electric field intensity in our experiment is about 2 V m^{-1} . However, the interaction time between the MV and the OM in our experiment was 20 minutes. Due to the limitation of simulation calculation ability, the actual calculation only simulates 30 ns. In order to speed up the calculation time of the interaction between the MV and OM, we increased the MV intensity to 1.0 V nm^{-1} . The high background electric field is typically used in MD simulations to probe poration³³....."

The valuable literatures have been cited in page **41** of the revised manuscript, as following (the numbers are their locations in the reference list in the manuscript):

"... 32. F. Apollonio, *et al.* Mixed Quantum-Classical Methods for Molecular Simulations of Biochemical Reactions With Microwave Fields: The Case Study of Myoglobin. *IEEE T. Microw. Theory.* **56** (11), 2511-2519 (2008).

33. J. Song, *et al.* Synergistic effects of local temperature enhancements on cellular responses in the context of highintensity, ultrashort electric pulses. *Med. Biol. Eng. Comput.* **49** (6), 713–718 (2011)....." in the manuscript.

Comment 2: As shown in Fig.2, there existed some holes in such a small system under the action of a very strong electric field. It is can be concluded that the membrane structure became looser and more porous after microwave irradiation. However, it is not reasonable to identify the holes as nanochannels. The authors should give verification and evidence of the existence of a channel.

Reply: Thank you very much for the professional suggestion. We are very sorry for our negligence of confusing the basic concepts of nanochannels and holes. Indeed, the outer membrane becomes loose and more porous after microwave irradiation. Following your advice, we have replaced "nano channels" with "nanopores" in the revised paper.

Fig. 5 e, SEM images representing the morphologies and structures of *E. coli* before and after different treatments. Scale bar, 2 μm .

As described in **Fig. 5e**, the "No obvious morphological changes were observed in the MV group ...Although MV produced instant channels in the OM of *E. coli*, the inner membrane of *E. coli* was intact and the channels in the OM disappeared when the MV irradiation stopped." Proving the existence of such instantaneous holes requires in-situ microwave irradiation to capture photos (nano-level precision). At present, we have no suitable strategies to complete it. Notably, we have demonstrated the existence of these nanopores through some indirect experiments. These experiments included the use of ANS probes to investigate changes in the permeability of bacterial membranes after application of microwaves (**Supplementary Fig. 19**); the use of confocal lasers to observe the apparent increase in bacterial permeability after application of microwaves (**Fig. 6**). These results are strong evidence that MV can generate nanopores in OM.

Comment 3: Why didn't the authors perform the MD simulation for the system including membrane and nanoparticles under microwave irradiation. It would be more pertinent to this study.

Reply: Thank you very much for the professional suggestion. As your suggested, we had supplemented the MD simulation for the system including membrane and nanoparticles under MV irradiation. First, we determined that the stable structure of GNs is GNs-1 by MD simulation (**Supplementary Fig. 18**). Thence, GNs-1 was used for subsequent transmembrane simulation experiments (**Fig. 5 g-i**). Additionally, in order to accelerate the membrane penetration process of GNs-1, we also supplemented the tensile dynamics simulation of the membrane and nanoparticles (OM-GNs-1) system with (**Fig. 5 j, k**) or without (**Supplementary Fig. 19**) MV. These results indicate that GNs can pass through the OM barrier and enter the cell through the pores generated by MV to achieve their bactericidal effect.

Supplementary Fig. 18 GNs stability analysis. **a**, Representative configurations of molecular dynamics simulation of GNs in top 3 terms by molecular docking at 25, 50, 75, and 100 ns. Atomic color coding in crystal structure: C-cyan, O-red, and H-light grey. **b**, **c**, RMSD (**b**) and Rg (**c**) of these models in top three terms to evaluate their stability by MD simulation (100 ns). **d**, Specific model of GNs-1 by hydrogen bonds (red lines) and π - π interactions (yellow lines).

Fig. 5 g, The conformational change of the OM-GNs-1 system after MV application at 0 ns, 2 ns, 4 ns, 10 ns, 20 ns and 30 ns. **h**, Two-dimensional graph of density evolution of OM-GNs-1 system after MV application at 0 ns and 30 ns. **i**, Two-dimensional graph of the average density of the OM and GNs-1 in the equilibrium phase (10-30 ns) after applied

MV in OM-GNs-1 system.

Fig. 5 j, Dynamic behavior of GNs-1 during tensile dynamics simulation under MV. **k**, Displacement diagram of GNs-1 during tensile dynamics simulation.

Supplementary Fig. 19 Dynamic behavior of GNs-1 during tensile dynamics simulation without MV.

To better illustrate this point, we have modified the manuscript, as followed:

In Page 16, we added **Fig. 5 g-k**.

In Page S-19, we added **Supplementary Fig. 18**.

In Page S-21, we added **Supplementary Fig. 19**.

And, we have modified the manuscript, as followed:

In Page **S-19, S-20**, we added the statement “...We selected five monomers with higher content in GNs as the main components for constructing nanoparticles, and the number of monomer molecules in

GNS was based on their mass ratio (α -Gambogic acid 10, Gambogenic acid 7, Isogambogenin 2, Isomorellic acid 2, Allogambogic acid, and Desoxygambogenin 1). Three small-molecule dispersion systems of GNs (GNs-1, GNs-2, and GNs-3) were constructed using the Packmol program. To study the stability of GNs in solvent systems, 100 ns molecular dynamics simulations were performed for these three GNs. As shown in **Supplementary Fig. 18 a**, during the 100 ns simulation, the structure of GNs-1 remained compact and stable. On the contrary, the conformational changes of GNs-2 and GNs-3 are more obvious, and the sharpening of the surface has the risk of small molecules falling off (100 ns). In addition, the RMSD values (**Supplementary Fig. 18 b**) of the three GNs fluctuated greatly in the first 30 ns of the simulation, which was due to the gradual migration and aggregation of the five monomers from the dispersed state, and the self-assembly to form nanoparticles. After 30 ns, the RMSD value fluctuated less and tended to equilibrium, indicating that the assembled nanoparticles tended to be stable in the solvent system. It is worth noting that the fluctuation of the RMSD value of GNs-1 after system equilibrium is less than 1 Å, while the fluctuation of GNs-2 and GNs-3 is close to 3 Å, indicating that the stability of GNS-1 is better than that of GNs-2 and GNs-3. Besides, Rg (**Supplementary Fig. 18 c**) also showed a similar trend. GNs-1 possessed definite stability due to their abundant hydrogen bonds (red lines), strong p-p interactions (dotted yellow lines), and hydrophobic interactions on alkene branches (**Supplementary Fig. 18 d**). In These results demonstrate that GNs-1 has the most stable structure for subsequent MV-assisted transmembrane studies.....”

In Page 19-20, we added the statement “... Furthermore, we simulated the process of GNs traversing the OM under the MV. First, the most stable structure of GNs was identified as GNs-1 by evaluating the stability parameters RDMS and Rg (**Supplementary Fig. 18**, the detailed discussion process was shown in supplementary information). Thus, GNs-1 was used for subsequent transmembrane simulation experiments. As shown in Fig. 5g, GNs-1 can be adsorbed in the nanopore generated by the MV, and with the extension of the simulation time, GNs-1 can migrate along the pore, and finally achieve drug delivery across the OM (**Supplementary Video 5**). Additionally, the density distribution of the OM-GNs-1 system before (0 ns) and after (30 ns) MV application shows that the system density distribution is uniform before MV application, and a low-density region appears in the system after MV application, which corresponds to nanopores (**Fig. 5h**). For ease of observation, we extracted the mean density maps of the OM and GNs-1 after MV application, respectively. As show in Fig. 5i, after MV irradiation, the OM has an obvious low-density area, and the position of GNs-1 is just in the low-density area of the OM, indicating that the GNs can counteract the barrier of OM through nanopores in the OM after MV application. In order to accelerate the penetrating dynamics of GNs-1 under MV, a traction force (1000 kJ mol⁻¹nm⁻²) was applied to GNs-1 for a tensile dynamics simulation of the OM-GNs-1 system. As shown in **Fig. 5j**, the OM under MV is destroyed during the kinetic process, resulting in the gradual formation of

pores in the OM, so that the GNs-1 can quickly complete the transmembrane under the traction force (**Supplementary Video 6**). Conversely, mere traction stretching was unable to achieve the transmembrane of GNs-1 (**Supplementary Fig. 19, Supplementary Video 7**). This is further illustrated by the relatively large displacement of GNs-1 under MV (**Fig. 5k**). Therefore, GNs-1 under MV can pass through the barrier of OM and enter into cells to achieve antibacterial effect.....”

In Page **30**, we added the statement “...In order to accelerate the membrane penetration process of GNs-1, the tensile dynamics simulation calculation of the OM-GNs-1 system with or without MV were carried out. During the kinetic simulation, a directional traction force was given to GNs-1 to accelerate its transmembrane behavior, and the traction force constant was set to $1000 \text{ kJ mol}^{-1} \cdot \text{nm}^{-2}$. The simulation results were analyzed with Gromacs 2018 and visualized with VMD.....”

In Page **31-32**, we added the statement “...**Evaluation of GNs stability.** In the molecular dynamics process, a GROMOS 54A7 force field was applied to the GNs complexes, while water molecules were built by the TIP3P model. After the small-molecule dispersion system is established, the energy optimization of the system is carried out by the steepest descent method and the conjugate gradient method. In order to adapt the system to the simulated environment, the constant temperature (NVT) ensemble equilibrium and the constant pressure (NPT) ensemble equilibrium were carried out. During the equilibrium process, the temperature coupling adopts the V-rescale method, and the thermal coupling time constant is 0.1 ps. Then the GNs were simulated for 100 ns using the leapfrog algorithm, the integration step was set to 2 fs, the long-range electrostatic interaction was processed by the PME algorithm, the short-range Coulomb cutoff radius was set to 1.2 nm, and the van der Waals cutoff radius was set to 1.2 nm. The system adopts periodic boundary conditions in all directions and uses the LINCS algorithm to constrain the bond lengths of small molecules. The simulation results were tracked with Gromacs 2018 and visualized with VMD.....”

Comment 4: The MARTINI force fields were developed based on the reproduction of partitioning free energies between polar and apolar phases for various chemical compounds. The addition of microwaves with rapidly changing electric fields would certainly affect the microscopic energy distributions and polarizability. Thus the implement of the force field to microwave action may need to modify or be validated. At least, the authors should have a discussion on the aspect.

Reply: Thank you very much for the professional suggestion. As you said, the MARTINI force field is developed based on the reproduction of the free energy distribution between the polar phase and the non-polar phase of various compounds.

However, the Martini force field is also a coarse-grain (CG, coarse-grain) force field for the molecular dynamics simulation of biomolecular systems. The coarse-grained mapping used by Martini

force field follows the following rules:

- (1) In general, four heavy atoms are replaced by one particle;
- (2) Two to three heavy atoms in a cyclic molecule are replaced by one particle;
- (3) Four water molecules are replaced by one particle, and three particles are used if polarization is considered.

Obviously, there are many assumptions behind the Martini model, the most important of which is to ignore some atomic degrees of freedom. As a result, the interaction between the particles is effective, and the energy profile is highly simplified, which greatly increases the sampling rate, but loses the description of the details. Therefore, it needs to be emphasized that for the energy profile, the processing method of the force field is not to sample as accurately as possible, but should be sampled as efficiently as possible. This is also the meaning of coarse-grained processing, which is intended to simulate the dynamics of large-scale atomic groups. At the same time, the shielding of the entire system in the Matini model is set to be uniform, and the same shielding constant makes the electrostatic interaction not very accurate. Similarly, when an electric field is added, the effects of interatomic interactions, conformational changes, and degrees of freedom changes within the particles are ignored, which reduces the influence of the force field on the energy distribution and polarizability of the system.

On the other hand, the all-atom force field considers all the atoms in the molecule and defines its parameters, such as OPLS-AA, AMBER, CHARMM, etc., and solves equations for all atoms during the dynamics simulation process, which reduces the calculation speed. The GROMOS 54a7 selected in this study is a classical joint atomic force field in GROMACS. It improves the calculation rate of the all-atom force field by ignoring some atoms in the molecule (non-polar hydrogen atoms) in the calculation process and integrating the interaction with adjacent atoms that form bonds with it. That is, the calculation process is accelerated while ensuring the accuracy of the description of atomic interactions. Based on the above considerations, we used this force field for our system.

To better express this point, we have modified the manuscript, as followed:

In page **28**, we added: “... The GROMOS 54a7 is a classical joint atomic force field in GROMACS. It improves the calculation rate of the all-atom force field by ignoring some atoms in the molecule (non-polar hydrogen atoms) in the calculation process and integrating the interaction with adjacent atoms that form bonds with it. That is, the calculation process is accelerated while ensuring the accuracy of the description of atomic interactions. Based on the above considerations, we used this force field for our system.....”

Comment 5: As mentioned in the references cited by the authors (e.g. Ref.11), the microwave irradiation certainly lead to the temperature increase for the treated system. How did the authors consider this problem in the MD simulation? What is reason to perform the simulation under NVT or NPT?

Reply: Thank you very much for the professional suggestion. Indeed, microwave irradiation will cause the temperature of the treated system to rise. With this in mind, based on previous experience and the conditions for killing Gram-negative bacteria in this manuscript, the temperature of 4 mg mL⁻¹ GNS is about 55°C after 15 minutes of microwave irradiation, we determined 55°C as the base temperature (control group) in our kinetic simulation process.

Fig. 3 e, MV thermal curves of different concentrations of GNs.

As shown in Fig. 3e, the treated system was slowly heated up after ten minutes of microwave irradiation, and the temperature almost stabilized, so NVT and NPT with constant temperature effects were considered in the selection of the simulation ensemble. Gibbs (Josiah Willard Gibbs. *Elementary Principles in Statistical Mechanics*, 1901.) believes that the NVT ensemble is the simplest form of stable distribution, and the average value obtained therefrom has the closest relationship with thermodynamics, so it is most suitable for finding the macroscopic properties of matter in equilibrium. Therefore, when we optimized the system, we first performed a 200 ps NVT dynamic balance. Then, after the NVT is balanced, considering that the system may undergo volume exchange at this temperature, which will cause the volume change of the simulation unit, we chose the isothermal and pressure ensemble NPT, which is closer to the actual situation of the experiment. Based on the above considerations, we first performed NVT optimization and then performed NPT during the dynamic simulation process.

Comment 6: The box size of the simulation system was set as 8×8×20 nm, and a hole with a diameter of 5 nm was formed in the membrane at the end of simulation. Will the size of the simulation system affect the simulation results and conclusions?

Reply: Thank you very much for the professional suggestion. The size of the simulation system has no effect on the simulation results. There is still a 3 nm thick film around the hole, and the simulation unit has a periodic boundary condition, that is to say, there are infinite identical simulation units around it to eliminate the influence of the boundary so that the size of the simulation system does not affect the simulation result.

Comment 7: In vivo experiments, the operating parameters of MW irradiation should be given, such as the microwave electric field strength/power and operating time.

Reply: Thank you very much for the professional suggestion. Following your advice, we refined the description and actual operation steps of the MV parameters in the in vivo experiment, and the corresponding descriptions have been shown in **Page 38**: “...Next, a microwave probe (8 cm in diameter, 2.45 GHz, 0.07 W cm⁻²) was used to irradiate the lungs of the mouse for 5 minutes. After an interval of 1 minute, which is to bring the temperature down to body temperature, a second irradiation (5 minutes) was performed. This repeated 3 times for a total of 15 minutes of irradiation. Multiple irradiations can avoid tissue damage caused by high temperature of the target tissue.....”

Comment 8: The authors should point out how they controlled the temperature of target tissues of the mice. Where is the target tissue of the mice? The temperature was controlled at 55 °C ? How to determine if the temperature was controlled at a desired value?

Reply: Thank you very much for the professional suggestion. Following your advice, we have supplemented the experimental procedure about to monitor the actual temperature of the mouse lungs during the treatment in **Supplementary Fig. 23**. Specifically, the mice were anesthetized after inhaling the aerosolized GNs and shaved the abdominal hair, and then cut the skin on the side of the mouse's abdomen. The purpose is to uncover the skin of the mouse abdomen after the target tissue is irradiated by a microwave probe to facilitate the thermal camera to record the temperature of the lungs. The operation process of microwave therapy is shown in **Supplementary Fig. 23c**. The actual temperature of the lungs in the mouse is shown in **Supplementary Fig. 23d**. With the extension of the microwave irradiation time, the temperature of the lungs gradually increased and reached 50°C within five minutes.

The target tissue is the lungs of mice.

The temperature did not reach 55°C in the in vivo experiment. The temperature of the lungs was 50°C after 5 minutes of microwave irradiation. In the microwave treatment of mice with pneumonia, first microwave irradiation for 5 minutes, with an interval of 1 minute (to reduce the temperature to body temperature), and then continue irradiation for 5 minutes. This operation is repeated 3 times for a total of

15 minutes. The purpose of such multiple irradiations is to avoid tissue damage caused by excessively high temperature of the target tissue.

Supplementary Fig. 23 Experimental procedures for detecting the temperature of mouse lungs during treatment. a-c, Experimental operation steps; d, Temperature changes in the lungs of mice during treatment.

To better express this point, we have modified the manuscript, as followed:

In Page S-27, we added **Supplementary Fig. 23**.

In page 24-25, we added “...First, we detected that the lung temperature of the mouse after inhaling atomized GNs reached 50°C for 5 minutes after MV irradiation (**Supplementary Fig. 23**). In order to avoid tissue damage caused by excessive temperature, we will irradiate repeatedly (5 min each time) during the treatment.....”

In Page S-27, we added **Supplementary Fig. 23**. And added “...Specifically, the mice were anesthetized after inhaling the aerosolized GNs and shaved the abdominal hair, and then cut the skin on the side of the mouse's abdomen. The purpose is to uncover the skin of the mouse abdomen after the target tissue is irradiated by a microwave probe to facilitate the thermal camera to record the temperature of the lungs. The operation process of microwave therapy is shown in **Supplementary Fig. 23a-c**. The actual temperature of the lungs in the mouse is shown in **Supplementary Fig. 23d**. With the extension of the microwave irradiation time, the temperature of the lungs gradually increased and reached 50°C within five minutes.....”

Comment 9: The treatment in Ref.11 was focus on MRSA-infected osteomyelitis. So, is 55 °C also the temperature needed for infected pneumonia concerned in this study? Why?

Reply: Thank you very much for the professional suggestion. 55°C is not the temperature required for the treatment of pneumonia in this experiment. As shown in **Supplementary Fig. 23d**, after five minutes of treatment, the lung temperature of the mice was nearly 50°C, but did not reach 55°C. In order to avoid irreversible damage to the lung tissue due to high temperature during the treatment process, we use multiple short-term treatments, that is, microwave irradiation for five minutes and a total of three

treatments. In the manuscript, 55°C is used as the temperature "limit" because our previous experience in microwave treatment of osteomyelitis (*Nat. Commun.* **11**, 4446 (2020)) found that 55°C will not cause damage to surrounding tissues in a short period of time (5 min).

To better express this point, we have modified the manuscript, as followed:

In Page **25**, we added "...And, the treatment temperature of mouse lungs did not reach 55°C, which further illustrates the importance of MV and GNs synergistically against *E. coli*....."

Comment 10: In Supplementary Fig.1, "the $g(r)$ decreased sharply in the long range...This suggests that some of the remote OM molecules have diffused beyond the monitoring distance". What is the monitoring distance? What is its physical implication? And what is its value?

Reply: Thank you very much for your kind reminder. We are very sorry for misleading you caused by the incorrectly expression of this sentence. In fact, what we want to express is that $g(r)$ drop sharply at a long distance indicates that the probability of OM molecules appearing at the corresponding distance becomes smaller.

The radial distribution function $g(r)$ is a very important characteristic physical quantity reflecting the microstructure of materials. The method of molecular dynamics calculation of $g(r)$ is:

$$g_{AB}(r) = \frac{1}{\rho_{AB} 4\pi r^2 dr} \frac{\sum_{t=1}^K \sum_{j=1}^{N_{AB}} \Delta N_{AB}(r \rightarrow r + dr)}{6t}$$

Where N_{AB} is the number of A and B atoms in the system (A and B can be atoms of the same type); ΔN_{AB} is the number of B (or A) atoms within the range from r to $r+dr$ from A (or B) atom; K is the total calculation time (number of steps); dr is the set distance difference; ρ_{AB} is the density of the system.

Fig 1S. Two-dimensional schematic diagram of the radial distribution function of particles.

$g(r)$ characterizes the average distance between molecules within the molecular chain of the system, and can be used to analyze the accumulation of molecules in the system. $g(r)$ drop sharply at a long distance indicates that the probability of OM molecules appearing at the corresponding distance becomes smaller and it does not spread beyond the monitoring distance. When calculating the OM radial distribution function, we select OM particles as the origin and the average OM particle number density of the system is $\rho=N/V$, then the local time average density at r from the origin is $\rho * g(r)$, which is simplified definition of a uniform isotropic system.

In short, the physical implication of r is the distance between the particles that may appear and the reference particles in the simulation system (**Fig. 1S**). We calculated the probability of OM particles appearing at different r , and found that the probability of OM particles appearing suddenly smaller when $r \geq 1$ nm than the initial value which may be because nanopores have already occurred at this distance. This also provides a side argument for the formation of nanopores.

Comment 11: In Supplementary Fig.1(b), all the values of RDF $g(r)$ at 30 ns were lower than those at 0 ns, and $g(r)$ converged to the minimum 1 at earlier time comparing with that at 0 ns. RDF denotes the ratio of the local density of average density of the system. Whether some molecules were not included in the calculation of RDF? please explain the reasons.

Reply: Thank you very much for the professional suggestion. All molecules are included in the RDF calculation.

Supplementary Fig. 1 Radial distribution function ($g(r)$) change curves of OM at the group of MV within 30 ns.

The reason why all the values of RDF $g(r)$ of 30 ns are lower than the value of 0 ns is that under the electric field, OM aggregates and forms holes, resulting in a smaller probability of his appearance at a certain r . In other words, not some molecules are not included in the calculation of RDF, but all molecules are included in the calculation of RDF. It is due to the change in the structure of the aggregate state that the probability of his appearance at a certain distance becomes lower, so $g(r)$ is less than 0 ns.

Comment 12: In Supplementary Fig.10, the difference in result of antibacterial test between the 0.125MIC MV group and the control group were not as large as that illustrated in Supplementary Fig.8 (26.06%), especially for those in the third line.

Reply: Thank you very much for the professional suggestion. As the reviewer mentioned, the antibacterial rate in the third column of the 0.125 MIC group is only 3.80%, which is obviously lower than the 26.06% shown in **Supplementary Fig.8 (Supplementary Fig. 9 in the revised Supplementary Information)**.

Supplementary Fig. 9 Statistics results of the antibacterial ability of GNs against *S. aureus*.

Supplementary Fig. 11 *S. aureus* after 16 hours of co-culture with different concentrations of GNs (Ctrl, 0.125 MIC, 0.25 MIC, 0.5 MIC, MIC, and 2 MIC) spread onto LB agar plates and incubated at 37 °C for 20 hours.

In **Supplementary Fig.10 (Supplementary Fig. 11 in the revised Supplementary Information)**, the antibacterial rates of the three parallel groups in 0.125 MIC group from top to bottom were 47.08%, 27.29% and 3.80%, respectively. The 26.06% in **Supplementary Fig.8 (Supplementary Fig. 11 in the revised Supplementary Information)** refers to the average of these three parallel groups, namely $(47.08\%+27.29\%+3.80\%)/3=26.06\%$

Response to Reviewer 2#

Original Comment:

This study entails investigating the use of Garcinia nanoparticles to eradicate Gram-negative bacteria using microwave assistance to induce nanochannels in the bacterial outer membrane to induce antibiotic entry. In vitro and in mouse models, the authors use this approach to demonstrate the use of microwave irradiation to markedly increase drug entry into cells. Strengths of this study include its use of this interesting and novel antibiotic-sparing approach to enable treatment of Gram-negative bacterial infections and inclusion of molecular, in vitro and mouse models to demonstrate effectiveness. Overall, the methods appeared to be sound for establishing the mechanism of bacterial killing by the nanoparticles.

Reply: We would like to thank the reviewer for his/her positive recommendation.

However, the manuscript could better establish clinical feasibility of this approach. For the in vivo models, it is unclear why the authors chose to investigate and only report data for *S. aureus* and *E. coli* co-infection as polymicrobial pneumonia is relatively uncommon.

Reply: Thank you very much for the professional suggestion. Generally, the reported rates for polymicrobial infection vary between 5.7% and 38.4% (*Eur. Respir. J.* 2006, 27, 795–800; *BMC Infect. Dis.* 2015, 15, 64; *Crit. Care*, 2011,15, R209). However, the true incidence is complicated to determine and probably underestimated due mainly to many cases going undetected, particularly in the outpatient setting, as the diagnostic yield is restricted by the sensitivity of currently available microbiologic tests and the ability to get certain types of clinical specimens (*Respirology*, 2016, 21, 65–75).

On the other hand, the outbreak of COVID-19 aggravates the severity of nosocomial bacterial spread, and bacterial infections also increase the mortality of COVID-19 (*Clin. Microbiol. Infect.* 2020, 26, 1622; *J. Am. Med. Assoc.* 2021, 4, 335). The latest clinical reports disclose that many COVID-19 patients die of secondary infections, including antibiotic-resistant bacterial infections, rather than the virus itself (*Lancet* 395,1054–1062 (2020)). That makes us aware of the severity of bacterial pneumonia, so we chose the bacterial mixed infection models.

To establish effectiveness of this approach against Gram-negative bacterial infections, which seems to be the primary objective of this manuscript, *E. coli* mono-infection should also be tested and this data reported (even if negative) as this could suggest additional bacterial or host mechanisms at play that are required for eradication of *E. coli* in the co-infection model.

Reply: Thank you very much for the professional suggestion. As the reviewer mentioned, we added *S. aureus* monoinfect pneumonia and *E. coli* monoinfect pneumonia to exclude the influence of additional bacteria or host mechanisms. Refer to the following reply to your **Comment 8** for details of the experimental results.

The organisms tested here are lab strains so it is unclear how the approach would fare against hospital-associated pathogens which may have multiple mechanisms of antibiotic resistance and other cell wall alterations.

Reply: Thank you very much for the professional suggestion. As the reviewer mentioned, we added the antibacterial effect of GNs against clinical *S. aureus* strains (**Supplementary Fig. 12**). And the antibacterial effect of MV-assisted GNs therapeutic strategy on hospital-associated *E. coli* (**Supplementary Fig. 15**) and multi-drug resistant bacteria *Klebsiella pneumoniae* (**Supplementary Fig. 16**). The hospital pathogen test report has been uploaded as supporting documents.

Experimental results show that the MIC of GNs to hospital-derived *S. aureus* is also 64 ppm. GNs with MV-assisted can eliminate 96.91% of hospital-derived *E. coli* and 98.36% of hospital-derived multi-drug resistant *K. pneumoniae*.

Supplementary Fig. 12 GNs efficiently eradicate clinical *S. aureus* strains. **a**, The MIC test of GNs on clinical *S. aureus*. **b**, GNs against clinical *S. aureus* tested by spread plate method. **c**, Clinical *S. aureus* strain counts calculated from spread-plate assays after GNs treated.

Supplementary Fig. 15 The efficacy of GNs and MV synergistically killing clinical *E. coli* strains. **a**, GNs against clinical *E. coli* under MV (MV+) or not (MV-) tested by spread plate method. **b**, Clinical *E. coli* strain counts calculated from spread-plate assays after treatment with GNs under MV excitation for 15 minutes or not. **c**, Statistics results of the antibacterial ability of the clinical *E. coli*.

Supplementary Fig. 16 The efficacy of GNs and MV synergistically killing clinical multi-drug resistant *K. pneumonia* strains. **a**, GNs against clinical multi-drug resistant *K. pneumonia* under MV (MV+) or not (MV-) tested by spread plate method. **b**, Clinical multi-drug resistant *K. pneumonia* strain counts calculated from spread-plate assays after treatment with GNs under MV excitation for 15 minutes or not. **c**, Statistics results of the antibacterial ability of the clinical multi-drug resistant *K. pneumonia*.

To better illustrate this point, we have modified the manuscript, as followed:

In Page **S-13**, we added **Supplementary Fig. 12**. And we have added: “... The antibacterial tests revealed that the MIC of GNs against clinical *S. aureus* is 64 ppm (**Supplementary Fig. 12a**). Furthermore, the antibacterial effect of GNs against clinical *S. aureus* strains was also confirmed by the spread-plate assay (**Supplementary Fig. 12b**). Specifically, the clinical *S. aureus* counts were obviously reduced to 0.002 (10^7 CFU mL⁻¹) in the GNs (MIC) group with 99.92% antibacterial rate (**Supplementary Fig. 12c**).”

In Page **S-16**, we added **Supplementary Fig. 15**. And we have added: “... GNs (4 mg mL⁻¹) has a weak antibacterial effect on *E. coli* (clinical). Notably, the antibacterial effect is significantly improved after synergistic MV treatment (**Supplementary Fig. 15a**). Specifically, *E. coli* (clinical) counts (10^7 CFU mL⁻¹) in the GNs + MV group average fell to 0.088 ($P < 0.0001$, compared to Ctrl), lower than other groups (**Supplementary Fig. 15b**), demonstrating that 96.91% *E. coli* (clinical) were killed by GNs during 15 minutes of MV treatment (**Supplementary Fig. 15c**). In contrast, the antibacterial rate of GNs and MV alone was 60.42% and 57.48%, respectively (**Supplementary Fig. 15c**), far lower than that of the group of GNs + MV.....”

In Page **S-17**, we added **Supplementary Fig. 16**. We have added: “... Similarly, GNs (4 mg mL⁻¹) has a weak antibacterial effect on multi-drug resistant *K. pneumonia* strains (clinical), and the antibacterial effect is significantly improved after synergistic microwave treatment (**Supplementary Fig. 16a**). Specifically, multi-drug resistant *K. pneumonia* strains (clinical) counts (10^7 CFU mL⁻¹) in the GNs + MV group average fell to 0.042, lower than other groups (**Supplementary Fig. 16b**), demonstrating that 98.36% *K. pneumonia* (clinical) were killed by GNs during 15 minutes of MV treatment (**Supplementary Fig. 16c**). In contrast, the antibacterial rate of GNs and MV alone was 36.35% and 36.72%, respectively (**Supplementary Fig. 16c**), far lower than that of the group of GNs + MV.....”

In Page **13**, We have added: “... It is worth noting that GNs are also highly effective in killing *S. aureus* from hospitals (**Supplementary Fig. 12**).”

In Page **17-18**, We have added: “... Notably, the MV-assisted GNs treatment method also has a significant bactericidal effect on hospital-derived *E. coli* (**Supplementary Fig. 15**) and multi-drug resistant *Klebsiella pneumoniae* (**Supplementary Fig. 16**).”

Finally, I am not an expert in these methods, a more robust exploration of potential collateral effects on human cells may also be warranted, particularly given the time required for microwave penetration to deep tissues.

Reply: Thank you very much for the professional suggestion. According to your suggestion, we added the effect of microwave irradiation (consistent with the irradiation conditions in animal experiments) on

the viability and morphology of A549 (alveolar epithelial cell) and L929 (fibroblasts). As show in **Supplementary Fig. 22**, after MV treatment for one day the viability of A549 and L929 cells decreased to 88.56% and 70.89%, respectively. And the cell viability was restored to 98.83% and 90.64%, respectively, after continuing the culture to the seventh day. Similarly, after MV treatment for one day, the morphology of A549 and L929 cells changed from a polygonal shape with filamentous pseudopodia to a fusiform antenna that shortened and shrank, and the cells shrank into a spherical structure. Continue to culture to the seventh day, the cell morphology returned to normal, and the number of cells increased. These results indicate that cell proliferation will be inhibited after MV treatment, but after a short period of recovery, the cells can gradually recover and begin to proliferate.

Supplementary Fig. 22 The effect of MV irradiation on the viability and morphology of normal cells. **a, c**, The viability of A549 (**a**) and L929 (**c**) were cultured for one and seven days after treatment of MV or not. $n = 6$ independent samples for **a** and **c**. **b, d**, Fluorescent images of A549 (**b**) and L929 (**d**) were cultured for one and seven days after treatment of MV or not. Scale bar, 50 μm .

To better illustrate this point, we have modified the manuscript, as followed:

In Page **S-25**, we added **Supplementary Fig. 22**.

In Page 24, We have added: "... Meanwhile, we verified that cell proliferation will be inhibited after MV treatment, but after a short period of recovery, the cells can gradually recover and begin to proliferate (Supplementary Fig. 22)."

In Page 36, We have added: "... For the MTT test method of A549 and L929 under MV irradiation, refer to the method of NIH 3T3. The difference is that after the cells adhered (after 24 h of culture), the MV was irradiated for 15 minutes. Other experimental steps remain unchanged...."

In Page S-25, S-26, We have added: "... Then we studied the effects of microwave irradiation on the proliferation and morphology of normal cells. As show in Supplementary Fig. 22, after MV treatment for one day the viability of A549 and L929 cells decreased to 88.56% (Supplementary Fig. 22 a) and 70.89% (Supplementary Fig. 22 b), respectively. And the cell viability was restored to 98.83% (Supplementary Fig. 22 a) and 90.64% (Supplementary Fig. 22b), respectively, after continuing the culture to the seventh day. Similarly, after MV treatment for one day, the morphology of A549 and L929 cells changed from a polygonal shape with filamentous pseudopodia to a fusiform antenna that shortened and shrank, and the cells shrank into a spherical structure (Supplementary Fig. 22b, d). Continue to culture to the seventh day, the cell morphology returned to normal, and the number of cells increased (Supplementary Fig. 22 b,d). These results indicate that cell proliferation will be inhibited after MV treatment, but after a short period of recovery, the cells can gradually recover and begin to proliferate."

There were also many errors in spelling and sentence structure and poor clarity of some sections of the manuscript that require further revision.

Reply: We are very sorry for our negligence and we have carefully revised our manuscript for spelling and sentence structure following your advice.

Comment 1: Lines 50-51: This is perhaps meant to refer to the discovery of new drug classes e.g. from screening synthetic chemical libraries; several new antibiotics particularly with activity against carbapenem-resistant and other multidrug-resistant Gram-negatives have been approved in the last few years.

Reply: Thank you very much for the professional suggestion. Indeed, as you said, the accurate expression of Lines 50-51 should be the discovery of a new class of antibiotics. A combination of scientific and economic factors has slowed the discovery and development of these life-saving molecules to the extent

that only six new classes of antibiotics have been approved in the past 20 years, none of which are active against Gram-negative bacteria (*Cell* **181**, 1518-1532.e1514 (2020)).

To better illustrate this point, we have modified the manuscript, as followed:

In **Page 3**, made a revision: "... and none of the new class of antibiotics existing for combating Gram-negative bacteria have been approved in the past 20 years...."

Comment 2: Lines 58-59: Please restate as the outer membrane presumably predates the use of antibiotics.

Reply: Thank you very much for the professional suggestion. In order to eliminate the ambiguity caused by this sentence, we replaced "Gram-negative bacteria have evolved an outer membrane (OM) barrier to protect themselves from drugs" with "Gram-negative bacteria have two cell membranes, and most small molecules are unable to traverse the outer membrane (OM) and accumulate inside the bacteria".

To better express this point, we have modified the manuscript, as followed:

In **page 3**, made a revision: "... Gram-negative bacteria have two cell membranes, and most small molecules are unable to traverse the outer membrane (OM) and accumulate inside the bacteria...."

Comment 3: Lines 62-64: This sentence is unclear, please restructure. Also it would be helpful to include a brief summary of prior research specifically related to the Garcinia nanoparticles used here.

Reply: We would like to thank the referee for this valuable comment, which offers us a better understanding of the Garcinia. We restructured this sentence and actually what we want to express is "Similarly, many herbal medicine nanoparticles composed of a variety of small molecules are also unable to pass through OM, resulting in ineffective elimination of Gram-negative bacteria.". Meanwhlie, according to your suggestion, we summarized the biological activity of Garcinia in the prior research.

On this basis, the manuscript has been modified as followed:

In **page 3-4**, we made a revision "... Similarly, many herbal medicine nanoparticles composed of a variety of small molecules are also unable to pass through OM, resulting in ineffective elimination of Gram-negative bacteria.". And added "...As one kind of herbal medicine, garcinia and related extracts have been reported to possess multiple biological activities such as antibacterial (*S. aureus*), anti-inflammatory, antitumoral and antilipidemic properties^{15,16}. However, garcinia and related extracts have no activity against *E. coli*¹⁷....."

Comment 4: Lines 94-100: Please correct several typographical errors and fragment sentences. Please briefly state conditions in which prior testing of normal tissues was done and duration of anticipated

treatment tolerance.

Reply: Thank you very much for the professional suggestion. We are very sorry for our negligence and we have carefully revised our manuscript following your advice. And we added the test conditions of the previous work and the duration of anticipated treatment tolerance.

On this basis, the manuscript has been modified as followed:

In **page 6**, made a revision: “... Considering that the magnetic field force in the MV field is several orders of magnitude smaller than the electric field force²⁴, we ignore the influence of the magnetic field in the MV-OM interaction.....”

In **page 6**, we supplemented a statement “... In our previous work¹¹, we found that MV continued to irradiate for 15 minutes to gradually increase the body temperature to 55°C without damaging normal tissues. We anticipated the interaction between MV and OM will be completed within 15 minutes.....”

Comment 5: Lines 188-189: I don't believe heating inside tissues is reported here. To properly penetrate the human chest wall, depths >12 mm would need to be evaluated; were increasingly higher temperatures seen with treatment times >15 minutes?

Reply: Thank you very much for the professional suggestion. Indeed, this is only in vitro experimental data. In vitro, we use pork of different thicknesses as a model of biological tissues. The MV probe heats the GNs solution through pork of different thicknesses to simulate the heating effect of GNs in vivo (Supplementary Fig. 7). In order to better test the effect of microwave heating in the body of GNs, we supplemented the heating of the lungs during the treatment (Supplementary Fig. 23). The temperature of the lungs was 50°C after 5 minutes of microwave irradiation. In the microwave treatment of mice with pneumonia, first microwave irradiation for 5 minutes, with an interval of 1 minute (to reduce the temperature to body temperature), and then continue irradiation for 5 minutes. This operation is repeated 3 times for a total of 15 minutes. The purpose of such multiple irradiations is to avoid tissue damage caused by excessively high temperature of the target tissue.

Supplementary Fig. 23 Experimental procedures for detecting the temperature of mouse lungs during treatment. a-c, Experimental operation steps; d, Temperature changes in the lungs of mice during treatment.

Supplementary Fig. 7 Infrared thermal images of GNs through 24 mm pork tissue under MV excitation.

According to the reviewer's suggestion, we increased the thickness of the pork to 24 mm, and the GNs solution can still be heated to 55°C (**Supplementary Fig. 7**). Furthermore, through the infrared thermal images of the front view, the top view, and the tissue next to the probe, it can be seen that the tissue temperature has not increased significantly (about 30°C). These results illustrated that the GNs solution can achieve desired MV thermal performance even at 24 mm penetration depth without causing significant heating inside the tissues.

The microwave heating effect of GNs is mainly determined by its polar structure. Specifically, it is the microwave heating effect produced by the dipole vibration of the GNs and the frictional collision between the molecules caused by the microwave. The temperature measured by the thermal camera is the result of the interaction between the microwave heating effect and the heat dissipation of the GNs, so whether the final temperature rises or not is closely related to the temperature and humidity of the experimental environment at that time (*ACS. Nano.* 2018, 12, 2201–2210).

In this experiment, at a room temperature of 22.5°C and a humidity of 35%, the temperature of GNs will not increase significantly after being irradiated with a 0.07 W cm⁻² MV for 15 minutes. This is the result of the dynamic equilibrium of the microwave heating and material heat dissipation process.

On this basis, the manuscript has been modified as followed:

In **Fig 3f**, we added the 24 mm group.

In Page **S-8**, we added **Supplementary Fig. 7**.

In page **11**, we made a revision "... the MV power and irradiation time (Fig. 3f). Under 0.2 W cm^{-2} MV irradiation,". "... the MV power and irradiation time, the GNs solution can achieve desired MV thermal performance even at 24 mm penetration depth without causing significant heating inside the tissues (**Supplementary Fig. 7**)" And added "...It is worth noting that when the microwave power is adjusted to 0.4 W cm^{-2} , GNs is heated to 55°C with 24 mm pork in only 4 min....."

In Page **S-8**, we added "... We increased the thickness of the pork to 24 mm, and the GNs can still be heated to 55°C (**Supplementary Fig. 7**). Furthermore, through the infrared thermal images of the front view, the top view, and the tissue next to the probe, it can be seen that the tissue temperature has not increased significantly (about 30°C). These results illustrated that the GNs solution can achieve desired MV thermal performance even at 24 mm penetration depth without causing significant heating inside the tissues....."

Comment 6: Lines 224-227: Was GN given in combination with antibiotics for the duration of the treatment period? This is a bit confusing, please clarify. Please also state the concentration of GN used in comparable serial passage experiments.

Reply: Thank you very much for the professional suggestion. Antibiotics were not used during GNs treatment. The antibiotic group in Fig. 4e is only used for comparison with GNs to show that GNs is less likely to develop resistance compared with antibiotics commonly used in clinical practice.

As we stated in the "Serial passaging assay to evolve resistance" experimental section, the concentration of GN_s is 32 ppm (0.5 MIC) in comparable serial passage experiments.

To better express this point, we have modified the manuscript, as followed:

In page **35**, We added: "... The 0.5 MIC of GNs, Gent, and Ofloxacin against *S. aureus* are 32 ppm, 12 ppm, and 2 ppm, respectively....."

Comment 7: Lines 356-357: Further explanation of finding of in vivo safety experiments would be helpful here. What parameters were used to compare tissue samples between the treatment and control groups? Did this follow exposure to GN only or also GN+MV?

Reply: Thank you very much for the professional suggestion. The parameters we use to compare the group of control and treatment mainly include: (1). In vitro: cell viability (cell compatibility) and hemolysis rate (blood compatibility). (2) In vivo: blood routine (WBC, Lymph, Mon, Gran, RBC, HCT,

MCV, MCH, RDW, and MPV), hepatic function (ALT, TP, and TBIL), renal function (BUN, CR, and UA) and tissue sections of major organs(heart, liver, spleen, lung, kidney).

In vitro and vivo safety evaluation refers to GNs.

To better express this point, we have modified the manuscript, as followed:

In Page **24**, made a revision: “... Next, we used the methyl thiazolyl tetrazolium (MTT) method to evaluate the cytotoxicity of GNs at different concentrations (**Supplementary Fig. 21a**) and tested their blood compatibility by mouse blood (**Supplementary Fig. 21b,c**). The results show that cell viability can still reach more than 80% even the concentration of GNs reached 16 MIC and will not cause hemolysis (MIC), indicating that GNs have excellent biocompatibility and blood compatibility. To evaluate the safety of GNs in vivo, the blood tests, Hepatic function, Renal function and histological analysis were performed. As shown in **Supplementary Fig. 21 d-f**, no significant difference was observed in the blood routine (WBC, Lymph, Mon, Gran, RBC, HCT, MCV, MCH, RDW, and MPV), hepatic function (ALT, TP, and TBIL), and renal function (BUN, CR, and UA) analysis between the control (without surgery) and GNs groups at a given dose. These results suggest that GNs have no appreciable toxicity and are safe for in vivo application, which was further supported by the hematoxylin and eosin (H&E) results of the internal heart, liver, spleen, lung, kidney (**Supplementary Fig. 21g**).....”

Comment 8: Lines 360-362 (and similar comparisons throughout this section): Was infection with *S. aureus* or (especially) *E. coli* alone investigated for GN+MV in the in vivo models? The authors could also consider including a control group treated with standard antibiotics with activity against these two bacterial pathogens to show that improvement seen with GN+MV is similar to that seen with traditional antibiotics.

Reply: Thank you very much for the professional suggestion. Follow your valuable suggestions we added *S. aureus* monoinfect pneumonia (**Supplementary Fig. 24**) and *E. coli* monoinfect pneumonia (**Supplementary Fig. 25**) to exclude the influence of additional bacteria or host mechanisms. In addition, we chose the broad-spectrum antibiotic gentamicin (Gent) as the antibiotic treatment group (positive control). And the experimental data of the antibiotic group is supplemented in **Fig. 7 and Supplementary Fig. 26**. These results reveal that the improvement of GN+MV to pneumonia caused by *S. aureus* and *E. coli* is similar to that of traditional antibiotics.

Supplementary Fig. 24. Antibacterial effects of GNs on *S.aureus* mono-infect pneumonia in vivo. **a**, H&E staining images of infected lung tissues after 7 days of treatment. Scale bars, 1 mm and 50 μ m (enlarged view). **b**, Wright-stained images of blood in mice with *S.aureus* mono-infect after one day of treatment. Scale bars, 20 μ m. **c-f**, IL-6 levels (**c**), amount of Gran (**d**) and amount of WBC (**e**) for 1 day in blood. **f**, Bacteria counts in the infected lung after one day with different treatments. Data are presented as mean \pm standard deviations from a representative experiment (n = 3 biologically independent samples). P values were analysed by two-way ANOVA with Tukey's multiple comparisons post hoc test.

Supplementary Fig. 25. Antibacterial effects of GNs on *E. coli* mono-infect pneumonia in vivo. **a**, H&E staining images of infected lung tissues after 7 days of treatment. Scale bars, 1 mm and 50 μ m (enlarged view). **b**, Wright-stained images of blood in mice with *E. coli* mono-infect after one day of treatment. Scale bars, 20 μ m. **c-f**, IL-6 levels (**c**), amount of Gran (**d**) and amount of WBC (**e**) for 1 day in blood. **f**, Bacteria counts in the infected lung after one day with different treatments. Data are presented as mean \pm standard deviations from a representative experiment ($n=3$ biologically independent samples). P values were analysed by two-way ANOVA with Tukey's multiple comparisons post hoc test.

Fig. 7 Antibacterial effects of GNs on *S.aureus* and *E. coli* co-infected pneumonia in vivo. **a**, Macroscopic images of lung tissue after 7 days of different treatments. **b**, Micro-CT of lung in mice with co-infected pneumonia after 7 days treatment. Scale bars, 5 mm. **c**, Wright-stained images of blood in mice with co-infected pneumonia after one day of treatment. Scale bars, 20 μm . **d**, H&E staining images of infected lung tissues after 7 days of treatment. Scale bars, 1 mm and 20 μm (enlarged view). **e-g**, IL-6 levels (**e**), amount of Gran (**f**) and amount of WBC (**g**) from 1 to 7 days in blood. **h**, Bacteria counts in the infected lung after one day with different treatments. Data are presented as mean \pm standard deviations from a representative experiment ($n = 5$ biologically independent samples). P values were analysed by two-way ANOVA with Tukey's multiple comparisons post hoc test.

Supplementary Fig. 26 H&E staining images of heart, liver, spleen, lung, and kidney after 7 days post treatment.
Scale bar, 50 μ m.

To better express this point, we have modified the manuscript, as followed:

We have added an antibiotic group (Gent) to each test result in **Fig. 7** and **Supplementary Fig. 26**.

In Page **S-28**, we added **Supplementary Fig. 24**.

In Page **S-30**, we added **Supplementary Fig. 25**.

In Page **26**, We have added: “... Moreover, the therapeutic effect of GNs+MV is similar to that of Gent.”; “... In addition, the GNs group has a significant therapeutic effect compared with the Ctrl group ($P = 0.0301$) in eliminate *S. aureus*, but it is not effective against *E. coli*.” And added: “... which was similar to the Gent treatment group. Moreover, the improvement effect of GNs+MV to *S. aureus* monoinfect pneumonia (**Supplementary Fig. 24**) and *E. coli* monoinfect pneumonia (**Supplementary Fig. 25**) is similar to that of traditional antibiotics.”

In Page **27**, made a revision: “... mice in the Ctrl group developed myocardial fibrinolysis accompanied by protein mucus exudation (indicated by red arrows); a large number of irregularly shaped vacuoles (indicated by green arrows) and hepatocyte swelling (indicated by blue arrows) were seen in the liver cells; Clusters of red blood cells are gathered in the splenic sinuses with protein mucus exudation; some loop epithelial cells and loop mesenchymal cells in the medulla of the kidney tissue show watery degeneration (indicated by purple arrows). In contrast, after treatment with GNs, these abnormalities related to *S. aureus* and *E. coli* induced organ injury was partially alleviated, especially for GNs+MV group and Gent groups, which achieved better therapeutic effects. These findings indicate that GNs+MV

achieves the same effect of treating pneumonia in mice as antibiotics by reducing the number of bacteria and reducing organ damage.”

In page **S-28, S-29**, We added: “... The panoramic view of the HE slice in the mouse model of *S. aureus* monoinfect pneumonia we can see that there is a large area of focal infiltration of pink mucus inflammatory cells, and the enlarged image clearly shows the proliferation of alveolar epithelial cells (**Supplementary Fig. 24a**). In contrast, there were no obvious symptoms of infection in the GN+MV and Gent groups. Similarly, the Wright stained blood samples in the treatment group significantly reduced the number of lymphocytes (**Supplementary Fig. 24b**). And the IL-6 (**Supplementary Fig. 24c**), Gran (**Supplementary Fig. 24d**), and WBC (**Supplementary Fig. 24e**) levels in the GNs +MV and Gent group were significantly lower than that in Ctrl group, indicating that the bacterial infection was restrained in the treatment groups. In addition, the number of *S. aureus* in the lungs was significantly reduced in the treatment groups (**Supplementary Fig. 24f**). Notably, the number of bacteria in the GN+MV group was lower than that in the antibiotic group. These results fully prove the excellent performance of GN+MV in the treatment of pneumonia caused by *S. aureus*.....”

In page **S-30, S-31**, We added: “... Similarly, a large area of focal infiltration of pink mucus inflammatory cells appeared in the panoramic view of the HE slice in the mouse model of *E. coli* monoinfect pneumonia (Ctrl). And the enlarged image in the group of Ctrl clearly shows the proliferation of alveolar epithelial cells (**Supplementary Fig. 25a**). In contrast, the lung consolidation decreased in the GN+MV and Gent groups. Meanwhile, the Wright stained blood samples in the treatment group significantly reduced the number of lymphocytes (**Supplementary Fig. 25b**). And , the amount of IL-6 had decreased in the treatment group (**Supplementary Fig. 25c**). Besides, Gran (**Supplementary Fig. 25d**) and WBC (**Supplementary Fig. 25e**) levels in the treatment group (GNs +MV and Gent) were significantly lower than that in Ctrl group, indicating that the bacterial infection was restrained in the treatment groups. In addition, the number of *E. coli* in the lungs was significantly reduced in the treatment groups (**Supplementary Fig. 25f**). Notably, the number of *E. coli* in GN+MV group is almost equal to that of Gent group. These findings indicate that the improvement of GN+MV to pneumonia caused by *E. coli* is similar to that of traditional antibiotics.....”

In Page **37-38**, We have added: “...and positive control antibiotic group (Gent).”, “... for *S.aureus* and *E. coli* co-infected pneumonia mode.....”, “... For monoinfect pneumonia mode, replace 20 microliters of mixed bacteria suspension with single bacteria suspension.”, and “...And, the Gent group was treated with Gent (7.5 mg per 15 mice)”

Comment 9: Lines 385-387: Day 1 reductions in WBCs in the treatment groups may not be meaningful, particularly given that they were higher in subsequent days. It is also unclear what effect GN treatment itself may have on inflammatory parameters as there was no uninfected group treated with GN.

Reply: Thank you very much for the professional suggestion. We believe that the reduction in the first day of WBC in the treatment groups are meaningful.

Supplementary Fig. 21 In vitro and vivo safety evaluation. **a**, The viability of NIH-3T3 cells cocultured with different concentrations of GNs for three days. **b**, Hemolysis images of different concentrations of GNs. **c**, Hemolytic efficiency of different concentrations of GNs. **d**, Parameters of complete blood tests of mice after post-injection of GNs at 7 days. **e**, Hepatic function (ALT, TP, and TBIL) of Ctrl and GNs groups on day 7. **f**, Renal function (BUN, CR, and UA) of Ctrl and GNs groups on day 7. **g**, Histological analysis of internal organs injury in the given dose GNs. Scale bar, 50 μ m. Data are presented as mean \pm SD from a representative experiment ($n = 5$ independent samples for **a**, $n = 3$ independent samples for **c-f**). The n.s. present $P > 0.05$, and P values were analysed by two-way ANOVA with Tukey's multiple comparisons post hoc test for **c**, and P values were analysed by two-way ANOVA with Sidak's multiple comparisons post hoc test for **e, f**. WBC, white blood cells; Lymph, number of lymphocytes; Mon, monocyte; Gran, granulocyte; RBC, red blood cells; HCT, hematocrit; MCV, mean red blood cell volume; MCH, mean erythrocyte hemoglobin; RDW, red cell distribution width.

red blood cell distribution width. MPV, mean platelet volume; ALT, alanine transaminase; TP, total protein; TBIL, total bilirubin; BUN, blood urea nitrogen; CR, creatinine; and TBIL, total bilirubin.

Fig. 2 S, Amount of WBC in complete blood tests of mice after post-injection of GNs or not at 7 days.

Supplementary Fig. 21 shows a series of tests to evaluate the safety of GNs in vivo and in vitro, including blood routine tests. For the convenience of observation, we extracted the WBC data in the blood of normal mice after atomization GNs from **Supplementary Fig. 21d**. As shown in **Fig. 2S**, the content of WBC in mice after atomization GNs is not significantly different from that of normal mice, which shows that GNs itself will not affect the content of WBC.

Therefore, the decrease in WBC on the first day is because the decrease in the number of inflammatory cells caused by the decrease in the number of bacteria is not the inflammatory response caused by the GNs itself.

The significant change in the WBC value after treatment one day was due to the treatment group inhibiting the acute bacterial infection. The increase in WBC values in the following days may be due to the fact that the mice with pneumonia have only been treated once, and the bacteria that have not been completely killed have passed through the blood circulation, caused more serious organ infections and sepsis after 7 days.

Lastly, we would like to thank the Editor and all the Reviewers again for their time and effort in helping us improve the quality of this manuscript. It is greatly appreciated. We hope that our responses are satisfactory and the revised manuscript could meet the standard of *Nature Communications*.

Reviewers' Comments:

Reviewer #1:

Remarks to the Author:

The authors have revised the manuscript and addressed my comments.

Reviewer #2:

Remarks to the Author:

The authors are commended for their thorough and careful revisions, which have improved the manuscript. In particular, it was reassuring to see results reported for *S. aureus* and *E. coli* mono-infections in support of other findings, as well as the equivalent response of clinical and laboratory strains to the experimental treatment. It also helpful to see no evidence of other end-organ damage in the mouse model and further in vitro evidence of cellular recovery after MV treatment as well as better delineation of the depth of cellular penetration. The additional findings better support this approach as a potentially viable therapeutic strategy. The overall readability of the manuscript is improved in some sections, although throughout the text small errors in grammar remain and further editing may be required.

REVIEWERS' COMMENTS

Reviewer #1 (Remarks to the Author):

The authors have revised the manuscript and addressed my comments.

Reply: We sincerely thank you for valuable comments on our manuscript. Those comments are very helpful for us to revise and improve our paper.

Reviewer #2 (Remarks to the Author):

The authors are commended for their thorough and careful revisions, which have improved the manuscript. In particular, it was reassuring to see results reported for *S. aureus* and *E. coli* mono-infections in support of other findings, as well as the equivalent response of clinical and laboratory strains to the experimental treatment. It also helpful to see no evidence of other end-organ damage in the mouse model and further in vitro evidence of cellular recovery after MV treatment as well as better delineation of the depth of cellular penetration. The additional findings better support this approach as a potentially viable therapeutic strategy. The overall readability of the manuscript is improved in some sections, although throughout the text small errors in grammar remain and further editing may be required.

Reply: We sincerely thank you for valuable comments on our manuscript. Those comments are very helpful for us to revise and improve our paper.